

# A Prototype Method for Diagnosing High Ice Water Content Probability Using Satellite Imager Data

Christopher R. Yost[2], Kristopher M. Bedka[1], Patrick Minnis[2], Louis Nguyen[1], J. Walter Strapp[3], Rabindra Palikonda[2], Konstantin Khlopenkov[2], Douglas Spangenberg[2], William L. Smith Jr.[1] Alain Protat[4], and Julien Delanoe[5]

[1]NASA Langley Research Center, Hampton, VA, 23681, USA
[2]Science Systems and Applications, Inc., Hampton, VA, 23666, USA
[3]Met Analytics Inc., Aurora, Ontario, Canada
[4]Australian Bureau of Meteorology, Melbourne, Australia
[5]Laboratoire Atmosphere, Milieux, et Observations Spatiales, Guyancourt, France

*Correspondence to*: Kristopher M. Bedka (kristopher.m.bedka@nasa.gov)

**Abstract.** Recent studies have found that flight through deep convective storms and ingestion of high mass concentrations of ice crystals, also known as high ice water content (HIWC), into aircraft engines can adversely impact aircraft engine performance. These aircraft engine icing events caused by HIWC have been documented during flight in weak reflectivity regions near convective updraft regions that do not appear threatening in onboard weather radar data. Three airborne field campaigns were conducted in 2014 and 2015 to better understand how HIWC is distributed in deep convection, both as a function of altitude and proximity to convective updraft regions, and to facilitate development of new methods for detecting HIWC conditions, in addition to many other research and regulatory goals. This paper describes a prototype method for detecting HIWC conditions using geostationary (GEO) satellite imager data coupled with in-situ total water content (TWC) observations collected during the flight campaigns. Three satellite-derived parameters were determined to be most useful for determining HIWC probability: 1) the horizontal proximity of the aircraft to the nearest overshooting convective updraft or textured anvil cloud, 2) tropopause-relative infrared brightness temperature, and 3) daytime-only cloud optical depth. Statistical fits between collocated TWC and GEO satellite



parameters were used to determine the membership functions for the fuzzy logic derivation of HIWC probability. The products were demonstrated using data from several campaign flights and validated using a subset of the satellite-aircraft collocation database. The daytime HIWC probability was found to agree quite well with TWC time trends and identified extreme TWC

5    events with high probability. Discrimination of HIWC was more challenging at night with IR-only information. The products show the greatest capability for discriminating TWC $\geq 0.5$ g m$^{-3}$. Product validation remains challenging due to vertical TWC uncertainties and the typically coarse spatio-temporal resolution of the GEO data.

## 1 Introduction

Recent studies have documented many events since the early 1990s where aircraft flight through deep convective storms and cirrus anvil outflow has resulted in jet engine power loss, loss of engine control, and/or engine damage events (Lawson et al., 1998; Mason et al., 2006; Bravin et al., 2015). Mason et al. (2006) reported that engine events occurred in seemingly

15    innocuous cloud regions with only light to moderate turbulence, infrequent lightning, and where the pilot's radar indicated green or weaker echoes (< approximately 30 dBZ), leading to their hypothesis that the aircraft were encountering high mass concentrations of small ice crystals associated with convective updrafts, and ice accretion in the engine by ingested ice crystals was likely the cause of the events. These encounters of high mass concentrations in low radar

20    reflectivity have been termed high ice water content (HIWC) events.

Subsequent meteorological analyses support that HIWC can be found within or near convective updrafts with a high concentration of small ice crystals that would not produce a strong radar reflectivity (Platt et al., 2011; Gayet et al., 2012). Confirmation of many of the


original hypotheses of Mason et al., (2006) was obtained in an exploratory in-situ flight campaign in HIWC conditions (Grandin et al., 2014). In more recent years, icing of aircraft air data probes (e.g. pitot airspeed indicators) has also been recognized to occur in the same type of cloud conditions (Duvivier, 2010). Though pilots are using the best available storm avoidance guidance during flight, the microphysical characteristics of HIWC events make this hazard difficult to identify and avoid.

There is currently no formal definition of HIWC, and levels and exposure distances required for ice accretion in engines have not yet been established. Individual industry researchers have typically chosen between 1 and 2 g m$^{-3}$ as their HIWC thresholds, while accepting that 0.5 g m$^{-3}$ does not appear to represent a threat. Here we choose a HIWC threshold of 1 g m$^{-3}$, in the belief that, pending more information from ice accretion researchers on the threat levels, a more conservative definition is prudent for a detection scheme.

Geostationary (GEO) satellite observations and derived products have been used to analyze past HIWC events and could potentially be used to develop HIWC detection products, assuming that HIWC events occur in a consistent set of conditions. Through analysis of GEO data from historical in-service engine icing events over Japan and Southeast Asia experienced by Boeing aircraft, Bravin et al. (2015) found that these events occurred predominantly in mesoscale convective systems with relatively large anvils. The aircraft typically traversed through a long section (196 km on the average) of cloud at or near the convective equilibrium level, also known as the level of neutral buoyancy, but also quite close (41 km on average) to the center of a local area with cloud tops at least 10 K colder than the equilibrium level. These anomalously cold clouds often occur within or near "overshooting cloud tops" (OTs) which indicate small updraft cores (< 20 km diameter) of sufficient strength to penetrate through the



anvil, typically located near to the equilibrium level. A 10-K temperature differential equates to more than a 2-km penetration of cloud top above anvil based on analysis of GEO imagery coincident with NASA CloudSat OT overpasses (Griffin et al., 2016). Bravin et al. (2015) showed that not only were the events close in location to such an OT, but also typically within an

hour of the time of its maximum intensity, and not necessarily correlated to the time of the overall storm peak intensity. One HIWC nowcasting approach has recently been published which seeks to maximize the HIWC event detection rate by identifying any ice cloud with moderate to high cloud optical depth (COD > 20, de Laat et al., 2017). Use of such liberal detection criteria essentially indicates where HIWC could not occur, but does not attempt to

capture the physical processes responsible for generating HIWC that seem to occur within or near storm updrafts according to the Bravin et al. study.

     The European High Altitude Ice Crystal (HAIC) and North American High Ice Water Content (HIWC) projects (Dezitter et al., 2013, Strapp et al., 2016) have conducted several flight campaigns in recent years to collect in-situ and remote sensing observations of ice water content

within deep convective clouds at various temperature levels ranging from -10° to -50° C, a temperature range encompassing the typical ambient icing environment for aircraft climb, descent, and cruise. The overarching goal of these campaigns is to better characterize the atmospheric environment that causes engine and air data probe failures that threaten aviation safety. The campaigns seek to 1) help assess a new aircraft mixed-phase/glaciated icing

certification envelope (Mazzawy and Strapp, 2007) contained in FAA Title 14 Code of Federal Regulations Part 33 Appendix D that recently became law, by collecting in-situ characterization measurements for comparison, 2) develop techniques for the aviation industry to detect and nowcast HIWC conditions for exit and avoidance, 3) provide data for specialized aviation



applications such as engine modeling, ground cloud simulation facilities, and the development of certification means of compliance and 4) take advantage of the unique data sets the campaigns will provide to support scientific research in the characterization of deep convection, fundamental cloud microphysics, cloud modeling, and the development of radar and satellite

cloud remote sensing products.

The primary goals of the HAIC and HIWC campaigns were to provide statistics on the $99^{th}$ percentile of total water content (TWC), and characterize other relevant cloud parameters such as ice particle size for regulatory purposes.  The high-frequency in-situ TWC observations collected within the HAIC and HIWC campaigns are invaluable for development of satellite-

based HIWC diagnostic products.  Prior to these campaigns, a large sample of consistent in-situ TWC data across diverse geographic regions had been lacking, which had prohibited HIWC diagnostic product development. GEO satellite imagers such as the Multifunction Transport Satellite (MTSAT) Japanese Meteorological Imager (JAMI), Geostationary Operational Environmental Satellite (GOES), and Meteosat Second Generation Spinning Enhanced Visible

and Infrared Imager (SEVIRI) observed deep convective clouds throughout their lifecycles in 5 to 30 minute intervals during the HAIC and HIWC campaigns, depending on the geographic region. GOES-14 observations were also collected at up to 1-min intervals on two flight days during a Super Rapid Scan Operations for GOES-R (SRSOR, Schmit et al., 2014) period.  GEO imager observations and associated derived products depict storm intensity, locations of intense

updrafts, and cloud-top microphysical characteristics that have previously been analyzed for a small sample of HIWC events. A more comprehensive analysis of GEO imagery relative to aircraft TWC measurements enables the research community to better understand the



characteristics and evolution of clouds that do and do not produce HIWC. This understanding is critical for the development of satellite-based HIWC detection products.

This paper has two primary objectives, 1) to provide statistical analyses of cloud properties derived from GEO observations as a function of TWC using data from three recent flight campaigns and 2) to develop and demonstrate a prototype satellite-based probability of HIWC (PHIWC) diagnostic product. The three campaigns analyzed in this paper were the two HAIC-HIWC campaigns centered in Darwin, Australia in January-March 2014 and Cayenne, French Guiana in May 2015, and the NASA HIWC-RADAR campaign centered in Fort Lauderdale, Florida in August 2015. The prototype algorithm developed here provides a more precise method for near-real time avoidance of HIWC conditions than previously available.

## 2 Datasets

### 2.1 Experiment and Flight Campaign Descriptions

In 2006, an industry working group tasked by the FAA to review issues related to mixed-phase and glaciated icing conditions recommended the collection of a new in-situ data set to characterize the microphysical properties of deep convective clouds, the type that had been identified as causing engine events. The HIWC Study was then initiated within North America and initially worked on the modification and development of new instrumentation required to make accurate measurements in hostile convective conditions, while plans for a flight campaign were developed.

In 2012, the HIWC team partnered with the HAIC Project to conduct the first HAIC-HIWC flight campaign (hereafter Darwin-2014) in Darwin, Australia using the SAFIRE Falcon-20 aircraft. The flight program was successfully conducted between 16 Jan. and 18 Feb. 2014,





but fell short of its data collection goal due to an engine failure unrelated to engine icing. A total of 23 flights were conducted, mostly in large tropical mesoscale convective systems (MCSs). In addition to a new IKP2 TWC measurement device (described in the next section), the Falcon-20 cloud measurement instrument suite included a Droplet Measurement Technologies (DMT)

Cloud Droplet Probe (CDP-2; 2-49 µm), a Stratton Park Engineering Co. 2D-S imaging probe (10-1280 µm), and a DMT Precipitation Imaging Probe (PIP; 100-6400 µm). A Science Engineering Associates Robust hot-wire probe (Strapp et al., 2008, Grandin et al., 2014) provided backup TWC measurements. The RASTA (RAdar SysTem Airborne) research W-band radar (Protat et al., 2004) provided Doppler and reflectivity radar measurements from 6

antennae.

The second HAIC-HIWC flight campaign (hereafter Cayenne-2015) was conducted from 5-29 May 2015 out of Cayenne, French Guiana. Two aircraft collected cloud in-situ data from this campaign: the Falcon-20 with approximately the same instrument suite as described above, and the National Research Council of Canada Convair-580 aircraft, also equipped for for cloud

in-situ and remote sensing. The details of the many Convair instruments will not be provided here, but it carried a second copy of the IKP2 probe for primary TWC measurements, and the CDP-2, 2D-S, and PIP for particle size distribution measurements. A total of 17 Falcon-20 and 10 Convair-580 flights provided data from oceanic MCS and continental storms that were suitable for the HIWC cloud analysis.

The NASA HIWC-RADAR flight campaign (hereafter Florida-2015) was conducted from 12-28 August 2015, with a base of operations in Fort Lauderdale, Florida. The NASA DC-8 aircraft was equipped with the IKP2 TWC probe, CDP-2, 2D-S, and PIP as primary cloud in-situ instrumentation. A total of 10 flights were conducted off the southeast coast of USA, the



Gulf of Mexico, and over the Caribbean. Although the primary objective of this campaign was to test pilot radar technologies, the project adopted the same cloud measurement strategies, focusing on oceanic MCS. The measurements included 4 flights in tropical storms Danny and Erika.

The overall objective of the three flight campaigns was to collect data to support the aviation industry objectives, and thus the flights were designed and funded with the first priority to collect cloud characterization data in the types of clouds that cause engine events, and in regions of those clouds that would be similar to what a commercial aircraft pilot might traverse. These objectives were not necessarily compatible with collection of the best scientific data. The

flight plans, described in the HIWC Science and Technical Plan (Strapp et al., 2016), were adopted for all three campaigns. The plans call for quasi straight-and-level runs across convectively active regions of tropical oceanic MCS clouds, similar to those described by Grzych and Mason (2011). These runs were to be aligned with regions of expected heavy rain below the aircraft, but through at most low reflectivity "green" echoes on the pilot's radar. The

pilot was given the discretion to optimize the run using his radar, and set up a pattern of reciprocal runs to survey this suspected area of HIWC. If regions of high radar reflectivity were observed at flight altitude, the pilot was instructed to traverse the system at a conventional safe proximity from the reflectivity core. Regions of lightning were to be avoided. The objective was to collect at least 100 32.2 km data segments (a regulatory reference distance) at each of the

4 primary temperature layers -50, -40, -30, and -10 C, ±5 C (in order of priority). The combination of the three projects achieved the data collection objectives, with a total of more than 54800 km of in-cloud data in mostly tropical oceanic MCS.





## 2.2 In-situ Total Water Content Measurements

The measurements of these flight campaigns were expected to be dominated by ice crystals in concentrations that could, at least in theory, reach 8-9 g m$^{-3}$ in the unlikely event that deep adiabatic cores were encountered. Although various methods have been used to make

airborne TWC estimates, including evaporators, these measurements have been especially problematic in a high altitude, high speed, high IWC environment, and lack substantive information on absolute accuracy. The aviation group tasked to develop the Technology Plan that recommended the flight measurement campaigns examined reliability and accuracy information for existing airborne TWC instruments and concluded that a new instrument was

required to reduce risk and provide more defensible accuracy. It was therefore decided to develop a new TWC instrument specially designed for this environment.

The new prototype isokinetic TWC evaporator (IKP1) was designed specifically for the high-altitude high-speed, high IWC environment, with a design goal of 20% accuracy at 10 g m$^{-3}$, 20 KPa, and 200 m s$^{-1}$ true airspeed (Davison et al., 2008, 2010). The probe was downsized to

a second version (IKP2) for use on the Falcon-20 aircraft, and then successfully tested in a series of wind tunnel experiments, exposing it to IWCs as high as 15 g m$^{-3}$ under high altitude flight conditions (Strapp et al., 2016). Accuracy estimates to date have been provided by Davison et al. (2016) and Strapp et al. (2016). System accuracy estimates predict no more than 10% error at all TWC values larger than about 0.1 g m$^{-3}$ for temperatures colder than -30°C. At warmer

temperatures, uncertainty rises due to the increasing magnitude of the background humidity, which must be subtracted from the IKP2 humidity signal. This produces a baseline uncertainty, limiting the practical use of the probe to about 0°C and colder, with the greatest effect on low TWC measurements. For the flight programs described herein, the IKP2 was the primary





instrument for the TWC measurements; it has so far been concluded that it provided high TWC measurements at the desired accuracy within the target -10 to -50 °C temperature intervals.

The measurements of the three campaigns have revealed that ice crystals dominated the TWC in the large deep convective clouds sampled. Mixed phase regions were narrow and

infrequent, and contained only low liquid water contents. In this regard, the terms high IWC and high TWC are synonymous for the purposes of this paper. HIWC will be used as a generic term to denote the condition, and high TWC will be used when referring to the measurement.

TWC values measured during the flight program reached a maximum of about 4.1 g m$^{-3}$ over a 0.93 km distance scale, the shortest scale length that is provided for this dataset.  Extended

periods of HIWC, defined as TWC $\geq$ 1.0 g m$^{-3}$, were not uncommon, with 92.6 km averaged TWC values reaching as high as 2.3 g m$^{-3}$.  An example of a flight pattern conducted during Darwin-2014 Flight 16 is shown in Fig. 1.  The IR image is color-enhanced such that the broad white area with embedded purple is at -78°C or colder, this temperature threshold being an overall informal estimate of the convective equilibrium level used during the project, and about

7° C warmer than the value estimated for this day. The WMO-defined (World Meteorological Organization, 1957) and cold point tropopause values were about -76 and -90° C respectively from the nearby Broome, Australia 8 Feb. 2014 00 UTC radiosonde. In this case, the distance the aircraft traversed through clouds defined by the white area of the image are unusually long for this project at about 140 Nm.  The peak TWC observed during this flight was approximately 2.5

g m$^{-3}$, and there were particularly long periods of sustained TWC greater than 1.5 g m$^{-3}$ (See Fig. 8c).

**2.3 Satellite Observations and Derived Products**



During the field campaigns, satellite data were ingested and analyzed in near-real time to provide mission planning support and post-mission studies. The data were utilized on site and distributed to various science team groups via the internet, where they remain available in digital and image formats (http://satcorps.larc.nasa.gov).

### 2.3.1 Satellite Imager Observations

Multispectral GEO satellite observations from the MTSAT-1R JAMI, and GOES-13 and GOES-14 Imagers are used to analyze convective cloud characteristics for a variety of observed IWC conditions and to produce the cloud property retrieval and overshooting convective cloud top (OT) detection products described below. Observations from the visible (VIS) and four infrared (IR) JAMI and GOES spectral channels are used to derive these products. These data were acquired from the University of Wisconsin Space Science and Engineering Center using the Man-computer Interactive Data Access System-X (McIDAS-X, Lazzarra et al., 1999). Table 1 summarizes the spectral channels, the spatial and temporal resolution of the observations, and the geographic boundaries of domains of interest for the HAIC-HIWC Darwin-2014, Cayenne-2015, and NASA HIWC-RADAR Florida-2015 flight campaigns. The Meteosat Second Generation (MSG) SEVIRI also observed the Cayenne-2015 campaign domain at 15-min intervals, but its data were not used in this study for the following reasons. 1) MSG observes this domain at a very oblique angle (61°) which can adversely impact cloud property retrieval accuracy and increase parallax errors and 2) the MSG 1 km VIS data required for daytime OT detection is unavailable for most of the daylight hours over this region.

### 2.3.2 Satellite-Derived Cloud Property Retrievals



Cloud properties are retrieved from 4-km resolution GOES and MTSAT JAMI imager radiances for pixels classified as cloudy using the Satellite ClOud and Radiation Property retrieval System (SatCORPS) that identifies cloudy pixels (Minnis et al., 2008a,b), retrieves the cloud properties (Minnis et al., 2008b, 2011) and estimates radiative fluxes from multispectral

satellite imagery. For daytime portions of the images, defined as solar zenith angle (SZA) ≤ 82°, the Visible Infrared Shortwave-Infrared Split-Window Technique (VISST) is used to retrieve cloud properties such as thermodynamic phase, cloud optical depth (COD), ice crystal or water droplet effective radius ($R_{eff}$), cloud height, pressure, and temperature, and geometric thickness. The Solar Infrared Split-Window Technique (SIST) is used to retrieve these parameters at night.

It is important to note that the SIST IR-only COD is limited to values of ~6 and therefore is insensitive to optical depth variations within optically thick deep convective cloud tops, so only daytime VISST COD is employed in our study. The 1-km VIS data provided by GOES and MTSAT were subsetted to 4-km to match the resolution of the IR channels. The cloud phase algorithm classifies a cloudy pixel as either "liquid" or "ice" based on the cloud-top temperature

and $R_{eff}$ information. Optically thick clouds containing both liquid and ice are generally classified as ice clouds since the current version of the retrieval algorithm is unable to separately classify mixed-phase clouds. Ice Water Path (IWP) is not directly retrieved by VISST but rather is a parameter derived from COD and $R_{eff}$ that is intended to represent the total amount of water within the depth of vertically homogeneous clouds classified as having ice tops. Cloud phase is

combined with aircraft air temperature observations to ensure that both satellite and aircraft were sampling glaciated clouds. Full descriptions of VISST and SIST are provided by Minnis et al. (2011). For ice clouds, the reflectance model based on severely roughened hexagonal ice columns (Yang et al. 2008) replaces the smooth crysal model used in Minnis et al. (2011).



The VISST COD retrieval is designed to translate the observed VIS reflectance into COD through the use of cloud microphysical models and knowledge of solar illumination and sensor viewing geometry. There can be significant spatial variability in VIS reflectance within deep convective cloud tops due to shadowing induced by vertical perturbations such as gravity

wave and OT signatures. This is especially true at high SZA when the sun-facing sides of the vertical perturbation are illuminated, enhancing VIS reflectance, while the other side is shadowed.  The dark shadowed regions will yield a lower COD even though they have very similar cloud microphysics at the 4–km pixel scale as the brighter regions.  We show later in this paper that high TWC is more likely in high COD regions, so low COD induced by shadowing

would adversely impact our daytime PHIWC diagnostic product. Therefore, we smooth the COD using a 5x5 pixel box, weighting the center of the box by 9, the inner three-pixel frame by 3, and the outer five-pixel frame by 1 to preserve locally bright clouds but also to reduce shadowing artifacts.  These weights were selected based upon empirical testing. Examples of the VIS reflectance, unsmoothed COD, and smoothed COD products are shown in Fig. 2.  In the

unsmoothed COD (Fig. 2e), OT regions and deep convection are identified by high COD but gravity waves and other textured/shadowed regions induce lower COD and what may be considered to be "noise".  In the smoothed COD (Fig. 2f), the noise is greatly reduced but the high COD in deep convective anvils is preserved, which is our intent.

### 2.3.3 Automated Deep Convection and Overshooting Cloud Top Detection Methods

Deep convective anvil clouds and embedded updraft (i.e. OT) regions are detected and analyzed within GEO imagery using several automated methods.  A commonly used method for



deep convective anvil detection is the brightness temperature difference (BTD) between the ~6.5 µm water vapor (WV) and ~10.7 µm window (BTW) channels (Schmetz et al., 1997; Martin et al., 2008). Comparisons of the BTD product with CloudSat Cloud Profiling Radar observations indicate that positive BTD values are effective for detection of deep convection, often with a

cloud vertical thickness exceeding 12 km (Young et al., 2012).  This method has not proven to be effective for differentiating overshooting cloud tops (OT) from deep convective anvils (Bedka et al., 2010; Setvak et al., 2013), but increasingly positive BTD values have been correlated with intense storms and a higher likelihood of lightning (Machado et al., 2009).

An automated satellite-based OT detection method has been used to identify the locations

of anvil clouds and deep convective updrafts within these clouds (Bedka and Khlopenkov, 2016). OTs often appear in satellite imagery as small clusters of pixels having cold BTW and enhanced texture in the VIS channel relative to the surrounding anvil, which has much more uniform BTW and smoother texture.  A set of statistical, spatial, frequency analyses were developed to identify clouds that could be within convection and to detect embedded BTW minima and textured

regions.  A set of OT candidate regions corresponding to local BTW minima are initially defined. The region around OT candidates is then analyzed to define the spatial extent of the anvil cloud.  The anvil cloud boundary is identified by a rapid BTW increase indicative of anvil edge or sharp BTW fluctuations indicative of breaks in the anvil between neighboring storms. OT candidate regions that pass an extensive set of tests that quantify how closely a candidate

resembles a typical OT are assigned a final "OT Probability" based on three parameters: the temperature differences between the OT candidate minimum BTW and 1) the surrounding anvil mean BTW, 2) the tropopause temperature derived from the WMO lapse-rate definition and 3) the most unstable equilibrium level temperature.  Parameters 2 and 3 are derived from the





Modern-Era Retrospective Analysis for Research and Applications data (MERRA, Rienecker et al., 2011), but these parameters could also be derived from any analysis or forecast for real-time applications. The OT Probability product was trained on a large sample of OT signatures and non-OT anvil cloud, with the intent to assign high probability to prominent OT signatures and

low probability to subtle within-anvil BTW minima that are less likely to be an OT. Of the three parameters listed above, the OT-anvil BTW difference has the greatest impact on OT Probability.

       The texture in VIS imagery is quantified via a unitless "VIS Rating". The VIS Rating product is based on pattern recognition within Fourier transform analyses of small windows

(32x32 ~1-km VIS pixels) of pixels with VIS reflectance consistent with optically thick anvils observed at a particular location and time of day/year. OTs and gravity waves produce a unique signature within the Fourier spectrum, and the prominence of this signature is quantified to derive the VIS Rating. Although VIS Rating values can exceed 50 for the most prominent OTs that penetrate the tropopause by 2+ km (Sandmael et al., 2017), values greater than 7 were found

within a majority of human-identified OTs (Bedka and Khlopenkov, 2016). Values as low as 5 identify enhanced cloud-top texture indicative of vertical motions within the cloud or gravity waves and possible generation of HIWC, but these low VIS Rating regions often do not correspond with the classic "cauliflower-like" signature that a human would consider to be an OT. The VIS imagery is processed at its original 1-km resolution within the OT texture

detection algorithm, but the maximum VIS Rating for each 4-km IR pixel region is recorded in the final VIS Rating product.

       The OT detection algorithm offered a 69% POD and 18% FAR when high probability (≥ 0.7) BTW-based OT detections were compared against a large sample of human OT



identifications within 0.25 km MODIS VIS imagery in Bedka and Khlopenkov (2016). These accuracy statistics are based on automated detections using MODIS imagery sampled to a 4-km resolution, representative of MTSAT JAMI and GOES imager data analyzed in this paper. The POD and FAR changed to 51% and 2%, respectively, when high OT Probability detections were

collocated with VIS Rating, illustrating that VIS texture detection can be used during daytime to confine the BTW-based product almost exclusively to OT regions. Areas with a non-zero VIS Rating outside of human-identified OT regions in GOES-14 imagery often coincided with regions of >30 dBZ radar echoes at a 10-km altitude for a case over the U.S., indicating the presence of strong updrafts (Bedka and Khlopenkov, 2016). In this paper, we will consider an

"OT or texture detection" to be an OT Probability $\geq 0.5$ or VIS Rating $\geq 5$ to attempt to capture all regions within or near strong vertical motions that generate detectable perturbations within the satellite-observed cloud top. Bedka and Khlopenkov (2016) provide a more comprehensive OT detection algorithm description and additional product examples.

    Within an anvil cirrus cloud, BTW is well correlated with the level of neutral buoyancy

temperature, which can vary regionally and seasonally (Takahashi and Luo, 2012; Bedka and Khlopenkov, 2016). Therefore, use of a fixed BTW threshold to discriminate HIWC conditions will not work across the globe so the BTW must be normalized. The computation of the level of neutral buoyancy is very sensitive to the boundary layer temperature and moisture profile, and reanalyses such as MERRA may not capture the actual boundary layer structure present at the

place and time of a satellite-observed storm. These inaccuracies can bias the derived level of neutral buoyancy and adversely impact BTW normalization. A more temporally and spatially stable reference for BTW is the tropopause temperature. The difference between BTW and the MERRA tropopause temperature (denoted as ΔBTW hereafter) provides a globally consistent



metric of storm intensity. Distributions of BTW and ΔBTW along aircraft tracks during the three

flight campaigns are shown in Fig. 3. We have defined ΔBTW as BTW minus the MERRA

tropopause temperature, so negative ΔBTW corresponds to cloud tops colder than the tropopause

temperature. Clouds most frequently sampled in Darwin-2014 had cloud tops approximately 25

K colder than those from Cayenne-2015 or Florida-2015 due to a higher mean level of neutral

buoyancy and tropopause in the Darwin region. Normalizing BTW by the tropopause

temperature shows that Darwin cloud tops were only about 5-10 K colder than Florida, but still

much colder (~20 K) than Cayenne.

Examples of MTSAT JAMI OT detection products for a specific time during a Darwin-

2014 campaign flight are shown in Fig. 2c. The corresponding VIS reflectance and ΔBTW

images (Fig. 2a-b) show numerous OT signatures present within ΔBTW ≤ 5 K in the center and

lower-left. Highly textured cloud without evidence of a classic "cauliflower-like" OT signature

is present in the upper right. Gravity waves are evident throughout the anvil cloud via ripples in

VIS texture emanating away from OT regions. Areas within and near OTs and other textured

regions are detected by the VIS Rating product (magenta contour, Fig. 2c). Especially cold and

distinct BTW regions collocated with VIS Rating detections are assigned high OT probability (>

0.7). Several other localized cold spots outside of textured regions in cold cirrus outflow and

gravity waves are assigned lower OT Probability. Bravin et al. (2015) concluded that HIWC

must often be present in outflow near to OT regions that are considered safe to traverse by pilots

due to the presence of weak or non-existent echoes from onboard pilot-radar systems. We

developed a distance-from-the-nearest OT (dOT) product (Fig. 2d) to quantify dOT-HIWC

relationships and verify the findings of Bravin et al. (2015) using a much larger sample size. The

dOT product will be discussed extensively in Sect. 3.



### 2.4 Satellite and Aircraft Collocations

Satellite observations, cloud property retrievals, and OT detections were collocated with

the aircraft TWC observations to characterize satellite-derived cloud conditions for a range of

TWC and to develop the PHIWC product. The 4-km nominal resolution of the satellite

observations is far too coarse to resolve details at the 0.93 km scale of aircraft TWC

observations.   The 5-second TWC observations were averaged to 45-second intervals, an

approximately 8-km distance based on a nominal aircraft cruising speed, in order to reduce

subpixel scale variability and derive values more representative of the area within a GEO imager

pixel. The maximum allowed time difference between the aircraft measurements and matching

satellite observations is equal to the temporal resolution of the imagery, listed in Table 1, with

Cayenne-2015 sampled at  the lowest frequency on average. The four nearest satellite pixels (i.e.,

a 2x2-pixel box) were matched to the mid-point of each of the 45-sec segments and the mean

cloud properties were computed within this box to account for uncertainty in cloud position

within the time window.  Parallax corrections were based on the retrieved cloud top height and

pixel location relative to GEO satellite nadir.

It is important to consider only the matches where the aircraft was physically located

within cloud, and not above cloud top or below cloud base. We consider a datapoint to be in-

cloud when TWC $\geq$ 0.1 g m$^{-3}$. In total, 5371 satellite-aircraft matched 45-sec data points within

cloud were derived from the 50 flights during the three flight campaigns.  4598 of the matches

occurred during daytime, defined by SZA $\leq$ 82°.  67% of the matches were used for PHIWC

product training and the remaining 33% were used for validation, described in Sect. 3.5. We

cluster our satellite datasets into three TWC categories with low, moderate, and high TWC

defined as 0.1 $\leq$ TWC < 0.5, 0.5 $\leq$ TWC < 1.0, and TWC $\geq$ 1.0 g m$^{-3}$, respectively, for discussion



purposes. TWCs of 0.5, 1.0, and 2.0 g m$^{-3}$ correspond to the 50$^{th}$, 75$^{th}$, and 95$^{th}$ percentiles, respectively, for this specific satellite-aircraft matched dataset. Note that the matched aircraft-satellite dataset discussed in this article is different from the dataset used for the HAIC-HIWC and NASA HIWC-RADAR regulatory analysis. The latter will supersede the dataset of this

article for any regulatory purposes.

## 3 Results

We begin our presentation of the results by showing comparisons of satellite observations, GEO satellite-derived products, and in-situ aircraft observations for a flight during

the Darwin-2014 campaign to demonstrate the set of satellite products most relevant for incorporation into the PHIWC diagnostic product. We then show how select satellite products relate to aircraft TWC using the 5371 matched data pairs. We follow with a description of the PHIWC diagnostic and conclude this section with PHIWC examples and validation.

### 3.1 GEO Satellite Products and In-Situ TWC Comparisons

Flight 22 of the Darwin-2014 campaign, the Falcon-20 took off from Darwin around 2145 UTC on 17 February 2014, sampled convective cells to the west over the Timor Sea, and returned to Darwin around 0100 UTC on 18 February. A time series of observations and derived products using the satellite-aircraft matched dataset for Flight 22 is shown in Fig. 4. Cloud top

heights were ~16 km for the duration of the flight corresponding to BTW around -80°C (Fig. 4a-b). TWC measurements were collected at two distinct flight levels, the first at -30°C from 2210-2345 UTC and the second at -40°C from 2350-0035 UTC (Fig. 4b) which were approximately 5-




6 km below our cloud top estimate. There were eight periods when TWC exceeded 1 g m$^{-3}$ and

two periods, 2230-2245 and 2315-2350 UTC, when TWC was relatively low (< 0.25 g m$^{-3}$).

TWC is not well correlated with BTW, ΔBTW, or BTD during this flight, indicating that

high TWC was present only in small regions within or beneath deep convective anvils (Fig. 4b-c,

e-f). BTW was consistently less than -80° C and ΔBTW near 0 K but the TWC varied

considerably with many periods of TWC near to or below 0.1 g m$^{-3}$. From 2315-0000 UTC, BTD

was greater than +1 K but varied by < 1 K as TWC increased steadily from ~0.01 to > 1 g m$^{-3}$.

This 1-K BTD variability is extremely small and within the noise of the MTSAT-1R JAMI WV

absorption channel (Ai et al., 2017), so the BTD is not useful for discriminating HIWC in this

case.

High TWC was observed in many instances when the aircraft flew close to OT detections

and in regions of high COD (Fig. 4g-h), showing that identification of small scale dynamical and

microphysical variability in broad and very cold cloud tops is critical for discriminating HIWC

conditions. The Falcon-20 was within 20 km of an OT detection from 2220-2230, 2255-2310,

2350-0005, and 0030-0035 UTC and sampled high TWC in all four encounters. TWC < 0.2 g m$^{-3}$ was observed during three other periods when the aircraft flew within 20 km of an OT (2321,

2342, and 2347 UTC). The COD was generally above 100 for all eight HIWC periods, but there

were short intervals when COD > 100 and TWC < 0.25 g m$^{-3}$, illustrating the challenge

associated with using passive GEO imager observations for isolating HIWC conditions at a

particular flight level. Sustained TWC minima around 2235 and 2315-2340 UTC correspond to

periods with the lowest COD (25-60) during the flight.  For reference, COD > 37 typically

indicates deep convection (Hong et al., 2007) and values exceeding 100 indicate extremely

optically thick cloud at or near deep convective cores.





### 3.2 Analysis of Cloud Properties as a Function of TWC

The analyses from Flight 22 on 17-18 February 2014 show that the BTD and ΔBTW products identify broad deep convective anvil cloud regions while dOT and COD resolve

smaller-scale structures within the anvil that are better correlated with TWC variability. Nevertheless, these results only represent one flight and it is important to examine how these and other parameters relate to TWC observed throughout the three flight campaigns. Fig. 5 shows CFDs of satellite-derived cloud properties for low, moderate, and high TWC conditions. To aid interpretation of the CFDs, we focus on Fig. 5c which features the clearest separation between

the three TWC intervals as a function of distance to the nearest OT pixel. This panel shows that 34% / 56% / 75% of low / moderate / high TWC events occurred within 10 km of an OT detection and 81% / 92% / 98 % occurred within 50 km of an OT.

There is a clear separation between the satellite-derived cloud properties for the low TWC and the two larger TWC intervals for all satellite parameters except $R_{eff}$ (Fig. 5e). Low

TWC values occur in warmer, less optically thick clouds that are farther from the nearest OT region than clouds with moderate or high TWC (Fig. 5a,d). IWP (Fig. 5f) is a parameter based on $R_{eff}$ and COD, so given that $R_{eff}$ provides no ability to discriminate between the TWC intervals, any separation between curves in the IWP plot is entirely driven by COD. Thus, despite the fact that de Laat et al. (2017) used IWP in their HIWC mask, we feel that IWP (as currently defined

for satellite remote sensing) appears to be a redundant covariant with COD for HIWC identification. Thus, IWP is excluded from further consideration in the algorithm formulation.

Differences between the moderate and high TWC categories are quite small for ΔBTW and BTD (Fig. 5b) but much greater for COD and dOT. High TWC occurred in flight within or



beneath anvils that were slightly (3 K) colder, a bit more characteristic of deep convection (0.2 K BTD increase), more optically thick (~15 COD units), and 6% more likely to be within 50 km of an OT region than moderate TWC events. Thus, it is clear that either moderate or high TWC can be present within or beneath deep convective cloud tops but high TWC occurs predominantly

5 near updraft cores, textured gravity wave regions, and optically thick and thus ice-laden cirrus outflow.

### 3.3 Probability of High Ice Water Content (PHIWC) Diagnostic Product Description

The fundamental goal of the PHIWC diagnostic product is to optimally combine a set of

10 satellite-derived parameters to assign high PHIWC to high TWC environments and much lower PHIWC to low and moderate TWC environments. The results from Fig. 5 show that our primary challenge will be differentiating moderate from high TWC, caused by the fact that GEO satellite imagers are most sensitive to cloud top and vertical integral parameters, leaving us to infer processes occurring within the cloud at flight level from temperature, reflectance, and spatial

15 patterns at the cloud top. The most promising parameters to include in the PHIWC model are COD and dOT. We also include ΔBTW to 1) address the fact that not every OT is accurately detected, so pixels with low ΔBTW far from the nearest OT can still achieve a relatively high PHIWC and 2) to provide additional information to the PHIWC model at night given that VIS-based predictors are unavailable. Previous studies (Bedka et al., 2010, 2012) and Fig. 5b show

20 that BTD does not offer unique information beyond that provided by ΔBTW. In addition, the spectral coverage of WV channels across the global constellation of GEO imagers differs slightly which causes differences in the observed BTs, resulting in inter-satellite inconsistencies in





PHIWC product output. For these reasons, BTD will not be considered further in the algorithm formulation.

Another way to view the relationships between COD, dOT, ΔBTW and TWC is in the form of scatterplots, shown in the left panels of Fig. 6. TWC increases sharply as a cloud reaches a height within 15 K of the tropopause, but then only increases slightly as the cloud top reaches or exceeds the tropopause height. There were many observations of low TWC for ΔBTW near 0 K, so cold cloud temperatures alone are an insufficient discriminator of HIWC. TWC also generally increases with increasing COD, but significant scatter is evident. Scatter may be due to lingering "noise" within the COD field due to shadowing/texture that we were unable to smooth as described in Sect. 2.3.2. TWC > 0.5 g m$^{-3}$ seldom occurs with COD < 37, the Hong et al. (2007) deep convection criterion. The best HIWC discriminator appears to be the VIS+IR dOT, with a concentration of TWC > 0.5 g m$^{-3}$ values at dOT < 10 km. A VIS rating ≥ 5 encompasses a much larger area than an IR OT detection which helps to explain the differences between the distributions for dOT with and without VIS information. Nevertheless it is extremely rare for more extreme TWC values (≥ 2.0 g m$^{-3}$) to occur outside of a VIS+IR dOT of 20 km, highlighting the importance of including detection of textured cloud tops in the PHIWC product.

For all parameters discussed here, a TWC threshold of 0.5 g m$^{-3}$ (50$^{th}$ percentile of TWC, vertical dashed line in Fig. 6) appears to be a breakpoint in the distributions where the satellite-based parameters begin to lose sensitivity. A useful PHIWC product should be able to discriminate regions with TWC ≥ 0.5 g m$^{-3}$. This TWC threshold is used to define a correct vs. false detection for the receiver operating characteristic (ROC) curve discussed in Sect. 3.5.



The right panels of Fig. 6 show boxplots of TWC as functions of ΔBTW, COD, and dOT. The boxes indicate the 25$^{th}$, 50$^{th}$ and 75$^{th}$ percentiles (Q1, Q2, and Q3, respectively) of TWC. The interquartile range IQR is given by Q3-Q1, and the lower and upper whiskers represent the extent of TWC values up to Q1 − 1.5*IQR and Q3 + 1.5*IQR. TWC values outside of this range are outliers and plotted as circles. Mean TWCs are indicated as solid black dots. Using fits to the mean TWCs (magenta curves), we derived the mean TWC across a set of ΔBTW, COD, and dOT intervals by inverting the axes on the scatterplots described above to derive a TWC "prediction" for a given parameter value (magenta curves, Fig. 6, right panels). These predictions are then combined to derive a final PHIWC for each satellite pixel using a fuzzy logic approach. Though it may not be clear in the scatterplot, TWC on average increases sharply as cloud tops approach the tropopause. Despite the scatter in COD, a linear relationship between the mean TWC and COD is evident. The mean TWC increases sharply within 40 km of an OT detection regardless of whether IR-only or VIS+IR OT detection is considered. The TWC increase when VIS dOT is included, however, is much sharper at a 0-40 km radius than the IR-only dOT. The dOT results are consistent with Bravin et al. (2015), who found that in-service engine powerloss events occurred on average 41 km from the center of locally colder and higher cloud regions embedded in the general cirrus canopy where OTs as described in this paper are often present.

We derive fuzzy logic membership functions for PHIWC based on the magenta curves in Fig. 6. The fits for ΔBTW or dOT have the exponential form shown in Eq. (1) whereas the COD is linear as shown in Eq. (2), with x corresponding to the ΔBTW, dOT, or COD value, and coefficients $c_1$-$c_3$ listed in Table 2.

$$TWC = c_1 \cdot c_2^x + c_3 \qquad (1)$$





$$TWC = c_1 \cdot x + c_2 \qquad (2)$$

We then linearly re-scale the TWC prediction values from Eq. (1) or Eq. (2) based on the TWC

derived from the Table 2 maximum ($TWC_{max}$) and minimum ($TWC_{min}$) PHIWC thresholds. The

net result is a PHIWC ranging from 0 to 1 for each parameter using Eq. (3):

$$PHIWC(\Delta BTW, dOT, COD) = \frac{TWC - TWC_{min}}{TWC_{max} - TWC_{min}} \qquad (3)$$

The PHIWC is set to 1.0 for parameter values at or above the maximum threshold. The

PHIWC($\Delta$BTW or COD) is set to 0 for values at or below the minimum threshold because the

matched TWC and satellite data show that it is extremely unlikely for HIWC to occur in such

warm or optically thin clouds. The PHIWC(dOT) is set to 0.01 for values at or below the

10   minimum threshold because the satellite imagery cannot resolve every OT nor can we detect

them all with automated algorithms. A PHIWC(dOT) of 0 would produce a final PHIWC of 0

using Eqs. (4-5) below which would be undesirable.

The final PHIWC value is derived by multiplying the individual parameter PHIWC

values together as shown in Eq. (4) for daytime imagery and Eq. (5) for night-time imagery:

$$PHIWC_{day} = PHIWC(\Delta BTW)^{w_1} \cdot PHIWC(dOT_{VIS+IR})^{w_2} \cdot PHIWC(COD)^{w_3} \qquad (4)$$

$$PHIWC_{night} = PHIWC(\Delta BTW)^{w_1} \cdot PHIWC(dOT_{IR-only})^{w_2} \qquad (5)$$

The weights, $w_1$, $w_2$, and $w_3$ defined in Table 2 were derived using an iterative approach to

20   maximize the area under the respective $PHIWC_{day}$ and $PHIWC_{night}$ ROC curves. $\Delta$BTW is the

highest weighted parameter for $PHIWC_{day}$ but adjusting the weights in favor of another

parameter did not appreciably decrease the area under the ROC curve (AUC, discussed in Sect.

3.5). For $PHIWC_{night}$, $\Delta$BTW has more weight than dOT. Overall, the difference in AUC for all

different weight combinations in the night and day products was less than 0.09.



### 3.4 PHIWC Product Examples

Examples of the PHIWC$_{night}$ and PHIWC$_{day}$ products valid at 2302 UTC (using the 2259 UTC MTSAT JAMI scan) during Flight 22 of the Darwin campaign discussed in Sect. 3.1 are

shown in Fig. 2g-h.  The aircraft-measured TWC within +/- 5 mins of the image time are overlaid on these graphs, showing seven high TWC (magenta X symbols) and other moderate TWC observations nearly coincident with this image. The Falcon-20 was flying northwestward toward an area of very cold cloud ($\Delta$BTW < 5 K).  The highest PHIWC$_{night}$ is concentrated near the coldest clouds and IR-only OT detections (Fig. 2c), as would be expected considering that

dOT IR-only and $\Delta$BTW datasets are used to derive PHIWC$_{night}$.   Several high TWC observations were located in ~20 K $\Delta$BTW and relatively far (~80 km) from an IR OT detection, combining to generate low (~0.3) PHIWC$_{night}$.  The VIS image (Fig. 2a) shows prominent gravity waves emanating away from the OTs to the northwest. Very optically thick cloud (COD > 100) was coincident with all high TWC observations.  Addition of these VIS-based products increased

the PHIWC$_{day}$ values to beyond 0.8. There does not seem to be any obvious reason in the satellite-derived products to explain why some moderate TWC observations were embedded within a sequence of high TWC observations, again highlighting the challenges in differentiating moderate from high TWC conditions using VIS and IR observations of cloud tops.

The PHIWC time series (Fig. 4d) for Flight 22 of Darwin-2014, described in Sect. 3.1,

shows that both the PHIWC$_{day}$ and PHIWC$_{night}$ products featured local maxima (> 0.8) at or near all periods (45 sec - 8 mins duration) when high TWC was observed.  These maxima were driven by COD peaks and flight through or near OT and/or textured regions, given that $\Delta$BTW was fairly constant throughout the flight. There were many other situations when flights within





optically thick clouds and low dOT measured low TWC.  As seen in Fig. 2f, areas of high COD can be quite broad and do not always pinpoint the high TWC observed at the -30 to -40° C flight levels.  The dOT product is included in PHIWC to depict the proximity to active convective cells that are more likely to generate HIWC conditions. It is very possible that the OT detection products are correctly detecting OTs when low TWC is observed, but the aircraft is upwind of the convective core and thus is not observing the high TWC that may be present downwind.  The PHIWC$_{day}$ and PHIWC$_{night}$ products were generally well correlated with each other except around 2256 UTC, a time very near to that highlighted in Fig. 2.  At 2256 UTC, the nearest IR-only OT detection was ~80 km away (Fig. 4g) with ΔBTW near 20 K which combine to reduce PHIWC$_{night}$.

An example of the PHIWC products and their inputs are shown in Fig. 7 for an image valid at 2312 UTC on 7 February during Flight 16 of Darwin-2014, also featured in Fig. 1.  The Falcon-20 flew very near to or within optically thick (COD > 90) and textured clouds with ΔBTW < 5 K ΔBTW. Texture and embedded BTW minima were detected well and the aircraft was frequently collocated with dOT < 20 km.  Unlike the scene shown in Fig. 2, high COD and texture, low dOT, and extremely cold cloud were collocated, leading to a similar appearance between PHIWC$_{day}$ and PHIWC$_{night}$.  High TWC was sustained for much of this flight segment and occurred within areas where PHIWC > 0.9.   PHIWC decreased slightly along the western edge of the segment in conjunction with a decrease to moderate TWC conditions.

A set of aircraft and satellite product time series for Flight 16 of Darwin-2014 are shown in Fig. 8.  The aircraft sampled clouds at -40° C which was about 6 km below cloud top.  There were ten individual high TWC encounters during this flight and HIWC conditions persisted for 7-10 min in many of the encounters.  PHIWC > 0.8 was present in nine of the ten encounters,


with the exception being a twilight high TWC encounter at 2204 UTC where ΔBTW of 13 K and

dOT IR-only of 40 km yielded a PHIWC of ~0.4. In general, the PHIWC time series was highly

correlated with TWC except for a high TWC encounter near 2245 UTC where lower COD (~50),

slightly warmer than average ΔBTW, and dOT > 20 km combined to produce ~0.6 PHIWC.

A segment from Falcon-20 Flight 19 of Cayenne-2015 shown in Fig. 9 illustrates an

especially challenging case for satellite-based HIWC detection.   The aircraft flew the long

segment shown at approximately -12 C through an anvil cloud much warmer than those featured

in Figs. 2 and 7.   The anvil was textured with a few embedded OTs and exhibited high COD.

High TWC was observed quite often during the 20-min segment.   The PHIWC$_{night}$ was extremely

low (~0.3) in high TWC regions due to the relatively warm cloud and lack of distinct BTW

minima and IR OT signatures.   However, the high COD and texture increased PHIWC$_{day}$ to

values over 0.75 throughout much of the region where high TWC was observed.   This shows the

value of VIS-based texture and COD products for capturing HIWC conditions in a cloud which

forecasters would not consider to be especially active based on BTW data and spatial patterns

alone.

          A set of aircraft and satellite product time series for Flight 19 of Cayenne-2015 is shown

in Fig. 10.   The aircraft sampled clouds at -12° C (7.1 km) during the first third of the flight and

at -44° C (11.7 km) for much of the remainder.   There were 3 (4) high TWC encounters at -12° C

(-44° C). When the aircraft sampled at -12° C, TWC was much higher and sustained, on average,

than during the rest of the flight. The corresponding values of PHIWC$_{day}$ were also greater,

mostly due to high COD (> 100). The PHIWC$_{day}$ during flight at -44° C was correlated with

TWC but the period of flight spanning 1720-1740 UTC was conducted near the northwest region

of the cloud where several new small quasi-isolated cells were observed, and consequently,



strong gradients in TWC and coincident cloud properties are evident in the time series around 1720, 1727, and 1735 UTC. Although the aircraft was very near OTs (dOT < 20 km) at these times, very low TWCs, ΔBTW > 30 K, and cloud top heights sometimes below flight level (Fig. 10a) all indicate that the aircraft exited the cloud for brief periods in this region. $PHIWC_{night}$ was

moderated by relatively high ΔBTW > 25 K even very near to OTs (dOT < 20 km) and did not exceed 0.70 throughout the entire flight. TWC of 1.4 g m$^{-3}$ is an extreme value for ΔBTW > 25 K and thus it is impossible to achieve a high $PHIWC_{night}$ in these conditions.  High TWCs observed at around 18:30 were associated with a different cloud system near Cayenne.

     Flight 5 on 19 August 2015 during the Florida-2015 campaign was observed by GOES-

14 during an SRSOR period, which provided images at 1-min intervals for almost the entire flight.  This high temporal resolution GOES-14 imagery enables precise matches between TWC and satellite products, virtually eliminating matching uncertainties, especially when compared to the 30-minute match window used for the GOES-13 data from the Cayenne-2015 campaign. The NASA DC-8 sampled a long-lived but gradually decaying MCS over Louisiana and the offshore

over the Gulf of Mexico. Time series of the observations and derived products for this flight are shown in Fig. 11.  The aircraft flew near the -50° C level for the first third of the flight and then ranged from the -30 to -50° C levels for the remainder. There were six high TWC periods observed during the flight and all of these periods were collocated with $PHIWC_{day} \geq 0.8$ and $PHIWC_{night} \geq 0.6$. Flight in dOT ranging from 0-20 km generally corresponded with periods of

greater TWC.  $PHIWC_{day}$ and $PHIWC_{night}$ were often consistent in magnitude.  The exception is the period from 1510-1540 UTC when the nearest IR-only OT was 30-60 km away but VIS texture was detected within 20 km.



Animations of aircraft TWC observations, satellite observations and products, and the PHIWC products are available at these links (PHIWC$_{day}$ https://cloudsgate2.larc.nasa.gov/prod/website/yost/2015231/4-panel_VIS+IR/, PHIWC$_{night}$ https://cloudsgate2.larc.nasa.gov/prod/website/yost/2015231/4-panel_IR/). These animation

shows that the PHIWC generally evolved quite smoothly due to a combination of the 1-min resolution of the imagery and the inclusion of ΔBTW and COD; high values of COD were generally persistent in time. The dOT product induces periodic flickering of high PHIWC that might be considered noise, especially during night when optical depth is not available to constrain the PHIWC product. But it is important to acknowledge that OT signatures can be

quite short-lived so some PHIWC temporal variability should be expected especially in the vicinity of pulsating updraft regions far removed from other updrafts. Additional animations for Flight 4 on 16 August observed by GOES-14 are also provided at these links to further demonstrate product behavior with this extremely valuable and rare 1-min resolution data. (PHIWC$_{day}$ https://cloudsgate2.larc.nasa.gov/prod/website/yost/2015228/4-panel_VIS+IR/,

PHIWC$_{night}$ https://cloudsgate2.larc.nasa.gov/prod/website/yost/2015228/4-panel_IR/)

### 3.5 PHIWC Product Validation

The PHIWC products were validated using a simple comparison of the probability of detection (POD) and false alarm rate (FAR) for varying PHIWC thresholds. 1771 night-time and

1571 daytime satellite-aircraft matches exluded from product training (see Sect. 2.4) were used for validation. As noted in Sect. 3.3, given that the satellite parameters show reduced sensitivity to TWC for moderate TWC values (> 0.5 g m$^{-3}$), we use 0.5 g m$^{-3}$ to define a correct detection but we also will discuss results relative to the more realistic HIWC TWC threshold of 1.0 g m$^{-3}$.





The ROC curves shown in Fig. 13 were constructed by plotting POD and FAR for thresholds chosen at intervals of 0.05 within the range 0-1.0. The thresholds are labeled at intervals of 0.1 along each curve. ROC curves using both 0.5 g m$^{-3}$ and 1.0 g m$^{-3}$ to define high TWC events requiring detection are shown in top and bottom panels, respectively, of Fig. 12. A

ROC curve for a perfect PHIWC product would intercept the (0.0, 1.0) coordinate and the area under the curve (AUC) would equal 1.0. The PHIWC threshold nearest the (0.0, 1.0) coordinate can be considered the optimal threshold because it yields the best compromise between POD and FAR. Therefore AUC was used as a metric to quantify the skill of the PHIWC product, and the optimal PHIWC threshold was identified based on maximum AUC. For PHIWC$_{day}$, (solid black

curve) the optimal PHIWC threshold is 0.70 and yields a 75% POD and 35% FAR based on the 0.5 g m$^{-3}$ threshold. This is consistent with the flight track time series results where a PHIWC$_{day}$ > 0.70 regularly identified high TWC events. The optimal threshold for PHIWC$_{night}$ (solid gray curve) is 0.55 and yields a 60% POD and 32% FAR. The PHIWC$_{night}$ POD would be 62% to achieve the same 35% FAR as the optimal PHIWC$_{day}$ product. FAR would decrease by up to

10% if a lower TWC threshold such as 0.01-0.05 g m$^{-3}$ were used to define cloud because the satellite-observed characteristics of clouds with such low TWC rarely triggers high PHIWC. But it is felt that 0.1 g m$^{-3}$ provides the most reliable detection of cloud boundaries and that statistics using this threshold are most representative of true product performance, keeping in mind the challenges associated with validation described below.

The reduction of skill for PHIWC$_{night}$ is not especially surprising given Fig. 6 that showed a relatively wide range of IR-only dOT and ΔBTW for TWC > 0.5 g m$^{-3}$. Lower PHIWC$_{night}$ values on average also reflect this uncertainty. The combination of these IR-based fields with COD and especially VIS texture detection more precisely depicts where HIWC is likely than the




IR-based fields alone. In the event that a COD retrieval product is unavailable due to latency

constraints, VIS OT and texture can be combined with IR OT detection to derive dOT and an

alternative two-parameter PHIWC$_{day}$ product. Based on TWC > 0.5 g m$^{-3}$, this alternative two-

parameter PHIWC$_{day}$ (Fig. 12, dashed black curve) would provide a 6% reduction in POD

relative to the three-parameter PHIWC$_{day}$ but a 7% improvement over the PHIWC$_{night}$ for a

constant 35% FAR

If HIWC conditions are defined as TWC > 1.0 g m$^{-3}$ rather than TWC > 0.5 g m$^{-3}$ the

accuracy of PHIWC is affected very little. The maximum AUC changes only slightly for all

three PHIWC versions. A higher PHIWC threshold is required to discriminate the 1.0 g m$^{-3}$

TWC events; for example the optimal PHIWC$_{day}$ threshold is 0.75 (versus 0.70 for 0.5 g m$^{-3}$)

which yields a 75% POD and 37% FAR.

A comparison of the PHIWC distribution for varying TWC intervals is shown in Fig. 13.

In general, both PHIWC$_{day}$ and PHIWC$_{night}$ increase as a function of TWC up to a value of 1.0 g

m$^{-3}$ but then level off at high PHIWC values in HIWC conditions. Very high TWC (> 2 g m$^{-3}$,

95$^{th}$ percentile in this 45-second dataset) seldom occurs when PHIWC$_{day}$ < 0.7 and PHIWC$_{night}$ <

0.4 during Florida-2015 and Darwin-2014 campaigns. Cayenne-2015 featured several cases of

low PHIWC$_{night}$ in very high TWC conditions, perhaps driven by the coarse 30-min GOES

sampling that cannot always capture rapidly-evolving OTs or cold cloud tops signifying HIWC

conditions. The Cayenne-2015 campaign featured the lowest PHIWC on average due to the fact

that clouds had the warmest cloud tops and greatest ΔBTW. There are many instances of low

PHIWC$_{night}$ but higher PHIWC$_{day}$ in HIWC conditions, further demonstrating the importance of

COD and VIS dOT. Challenges associated with PHIWC validation are discussed in the next

section that could help to explain perceived deficiences in the products.





One point to consider when interpreting these results is that 45-sec TWC data was used to develop and validate the TWC algorithms, not the native 5-sec data. Though 45-sec data is more representative of the size of a 2x2 GEO pixel clusters co-located with the aircraft than 5-sec, averaging across the 45-sec window dampens mesoscale variability that could contribute to

temperatures and spatial patterns observed by satellite, especially texture in 1-km VIS imagery that is 16 times finer than 4-km IR BT. For example, the PHIWC$_{day}$ threshold of 0.70 featured a 35% FAR based on 45-sec data. 41% of these false positives featured a 5-sec TWC > 0.5 g m$^{-3}$ and 1.0% featured TWC > 1.0 g m$^{-3}$. Of course the true positive rate would almost certainly decrease if, for example, 5-sec TWC > 0.5 g m$^{-3}$ required high PHIWC. Thus, there are both pros

and cons with the use of 5- versus 45-sec TWC data that complicate validation, but it is felt that the PHIWC algorithm based on 45-sec data is robust.

During the Darwin-2014 and Cayenne-2015 campaigns, the 95-GHz Doppler Radar System Airborne (RASTA, Protat et al., 2009) provided vertical TWC profiles above and below the aircraft that can be used to estimate if false detections from the PHIWC products based on

comparison with IKP2 were truly false, namely there were no retrievals of TWC ≥ 0.5 g m$^{-3}$ anywhere within the column. TWC observed during the flight campaigns decreased with height by 33% from the -10° C layer to the -30° to -50° C layer. If the aircraft was flying at -50° C and measured 0.4 g m$^{-3}$, TWC could exceed 0.5 or possibly 1.0 g m$^{-3}$ at lower altitudes in the same column. Moderate to high TWC at low to mid-levels could be correlated with cold and/or

textured cloud tops that would trigger high PHIWC values.     Flight-level RASTA TWC retrievals were found to have a ~10%–30% bias and 40%–70% root-mean-squared difference relative to in-situ TWC measurements during the Darwin-2014 campaign (Protat et al., 2016). Although RASTA TWC estimates remote from the aircraft level have not yet been validated for





accuracy, the radar TWC retrievals were considered to be adequate to analyze for the following false detections (Protat et al., 2016). A FAR of 31% was found for 0.7 $PHIWC_{day}$ for Darwin-2014 and Cayenne-2015 based on a 0.5 g m$^{-3}$ IKP2 threshold. 82% (35%) of the false detections were collocated with RASTA column-maximum TWCs that exceeded 0.5 (1.0) g m$^{-3}$ at heights

above the freezing level. This suggests that vertical sampling bias, especially when the aircraft flew at higher flight levels where TWC is lower on average, is likely influencing the IKP2-based validation statistics and therefore these statistics may not be truly representative of product performance.

On the other hand, when column-max RASTA TWC is used as truth for Darwin-2014

and Cayenne-2015, a nearly identical shape of the ROC curves relative to those from IKP2 (not shown) was found. When the column-maximum RASTA TWC is analyzed in conjunction with the satellite parameters used to derive PHIWC, very similar relationships are found with those shown in Fig. 6 which would lead to comparable RASTA-based PHIWC. So while some perhaps appreciable fraction of the false detection rate can be explained by vertical variability

and sampling bias, use of RASTA data is not enabling significant improvement in overall PHIWC accuracy.

## 4 Discussion

Our analysis found that HIWC conditions are most common during periods of flight

within or beneath optically thick cloud tops having temperatures near to or colder than the tropopause and within 40 km of an OT or textured cloud top. A combination of VIS-based texture detection and COD retrieval helps to pinpoint where HIWC conditions are likely within or beneath a broad area of cold anvil. High COD indicates a cloud top composed of dense ice



which generates high VIS reflectance. The $R_{eff}$ showed a nearly identical distribution in moderate and high TWC conditions, so the high COD and VIS reflectance is driven either by high ice mixing ratio in anvil clouds without deep convection underneath or by large vertical cloud thickness, i.e. the presence of deep convection. Convective updrafts depicted by OT

regions, high COD, and low ΔBTW are where strong ice mass flux occurs and the high TWC is likely be generated. The high TWC is then advected laterally within the anvil. These results are consistent with previous studies such as Grzych et al. (2015) and Bravin et al. (2015) who identified that engine icing events often occurred during long traverses through MCS clouds with cold tops, and in particular near to tops penetrating significantly above the surrounding anvil.

An alternative HIWC diagnostic method from de Laat et al. (2017) identifies any ice phase cloud top with moderate or greater COD (≥ 20) in the form of a binary HIWC mask. The goal of the de Laat et al method was to maximize the critical success index parameter which emphasizes POD. For example, based on a TWC threshold of 0.5 g m$^{-3}$, the de Laat method would have offered a 100% POD and 96% FAR using the daytime satellite-aircraft match

database from the three campaigns, indicating that this product is more useful for telling users where HIWC conditions cannot be present, rather than where it is present based on satellite-observed cloud conditions. It is unlikely that pilots would avoid an entire anvil simply because it is composed of ice and is at least somewhat optically thick. The approach described in this paper attempts to better discriminate where HIWC is present within a broad area of anvil which is

advantageous for tactical HIWC avoidance.

Validation of the PHIWC products using in situ TWC is a challenge and is potentially misleading for a variety of reasons. A primary reason for disagreement between PHIWC and TWC noted throughout this paper is the fact that the method is attempting to infer conditions at



flight level from observations of cloud tops. Consider the situation where an intense lightning-producing convective cell with an OT is producing extremely cold, optically thick, and spatially broad anvil that would generate a large area of high PHIWC. Low to mid-level cloud (flight level -10° to -30° C) containg low TWC may be present beneath and de-coupled from the anvil above.

This low TWC would agree poorly with the high PHIWC. Lightning detection data were used extensively in aircraft flight planning and routing during the campaigns. Active lightning-producing convective cores were intentionally avoided to ensure aircraft safety and preserve the science instrumentation, although the lightning measurements were also used to locate promising active cells for sampling shortly after the lightning had dissipated. In some flights, Flights 5 and

6 (19 and 21 August) in Florida-2015 for example, the aircraft could not sample freely due to the frequent combination of both lightning and high radar reflectivity at flight altitude. In such cases, the measurements were usually made along the edge or at an otherwise safe distance from the intense cell. This resulted in lower measured TWC values than would likely be present inside the cells. In vertically continuous clouds, a general decrease in observed TWC with height also

affects the validation statistics. Flight at -50° C may sample TWC below 0.5 or 1.0 g m$^{-3}$ but greater TWCs are likely to be present at lower flight levels in the same column. High PHIWC would be considered a false detection relative to TWC at -50° C scenario but correct at levels below.

The aircraft proximity to an updraft, downwind vs. upwind, may also have an impact on

observed TWC and thus our interpretation of satellite-derived data with respect to TWC. It is reasonable to assume that greater TWC would be sampled in new outflow downwind of an updraft versus upwind of the same updraft. New OTs often form in a cirrus shield composed of dissipating remnants from earlier decayed cells. The satellite could then observe cold, optically



thick cloud close to and all around an OT region, triggering high PHIWC, but the highest TWC

at given altitude would arguably be at a downwind position from the OT. It therefore stands to

reason that the winds derived from a numerical weather prediction (NWP) model analysis or

forecast could be another useful predictor for PHIWC. Unfortunately, outflow from deep

convection can alter the upper tropospheric wind environment. Models often do not simulate

convection at exactly the right place and time, and even if a storm were accurately simulated, the

model may not correctly simulate the interaction of the synoptic scale winds with the convective

outflow. These challenges would complicate use of the wind field as a PHIWC predictor in an

automated product. Another complication arises in an environment with multiple updrafts in

close proximity to each other, where the aircraft may be upwind of one updraft by several km but

downwind of another by tens of km. It would be difficult to understand exactly how, and from

where, the observed TWC is generated. Unraveling these complex relationships is a topic for

future work.

       Another point to consider when interpreting these results is that our analysis framework

treated each satellite image as an individual snapshot and did not account for the temporal

evolution of the clouds nor the distance which a commecial aircraft might traverse in high

PHIWC within the cloud. Both of these factors could offer additional value in a PHIWC

diagnostic product. Bravin et al. (2015) noted that in-service engine icing events occurred within

an hour of the maximum intensity of a local OT region that the aircraft traversed, supporting the

use of automated OT detection for identification of HIWC threat. Their analysis was based on a

manual analysis of storm system evolution and comparison to engine-event aircraft tracks. This

may be practical to do with an automated algorithm for large and isolated MCSs observed at high

temporal resolution, but it may be very difficult to identify OTs in shorter-lived smaller storms or





clusters of storms with anvil mergers. A primary contribution of uncertainty to the analysis of this article is caused by the coarse spatial and temporal resolution of the GOES and MTSAT imagers. These instruments cannot resolve all of the OT signatures that may truly be present in higher spatial resolution imagery such as that from the Moderate Resolution Imaging

Spectroradiometer (MODIS) and cannot depict the rapid temporal evolution of BTW and OT within updrafts. Exceptions to this were the two Florida-2015 flights sampled at 1-min intervals by GOES-14. One of these flights shown in Fig. 11 illustrates very good correlation between TWC and the derived PHIWC, suggesting that high temporal resolution imagery would improve our statistics. For example, if an OT had formed shortly after the first of two GOES images 30

minutes apart, it would not be registered on the first image. High TWC could then be observed by the research aircraft but associated with a low PHIWC value derived from the first image.

       Given the reduced $\text{PHIWC}_{night}$ performance relative to $\text{PHIWC}_{day}$, and in general, some error/uncertainty with the PHIWC products, consideration could be given to other parameters such as cloud top temperature and height available in near real-time that might improve

performance. In optically thick deep convection, the IR emissivity is very close or equal to 1, so the cloud top temperature is equivalent to the observed IR BT and therefore provides no new information. A cloud top height retrieval that properly handles semi-transparent clouds would offer value over IR BT, but this study has shown that high TWC almost never occurs within thin cirrus. Satellite observations in water vapor absorption channels have been used to derive night-

time COD which provide some sensitivity to COD variations in optically thick deep convective clouds (Minnis et al., 2016), much better than what is provided by the SIST algorithm described in Sect. 2.3.2 above that is limited to COD of ~8. The use of these new night-time COD retrievals will be explored in future PHIWC product versions.



Other data from NWP models such as thermondyamic parameters such as total precipitable water (TPW) or convective available potential energy (CAPE), or dynamic forcing for convection such as boundary layer convergence, mid-tropospheric (500 hPa) vorticity, or cloud top divergence, could also be included in a PHIWC product. High TPW is typically

present in broad areas of the tropics and would perhaps contribute to a greater concentration of water and ice hydrometeors in deep convection. It is very difficult to understand why high TWC was measured in some clouds and not others nearby in what appears to be same TPW environment based on NWP analyses. Complex mesoscale dynamics such entrainment of cold outflow and/or dry subsiding air from nearby convective cells would likely have a greater impact

on storm intensity and its ability to generate HIWC conditions than the TPW of the airmass depicted by a NWP model. The magnitude of CAPE is thought to govern the maximum updraft speed attainable in deep convection. Studies such as Rosenfeld et al. (2008) have suggested that stronger updrafts lead to generation of a greater concentration of small ice crystals. Given that previous studies have linked small ice crystals to HIWC events, it is possible that there would be

a statistical relationship between CAPE and high TWC. Greater CAPE and faster updrafts would also cause stronger ice mass flux that could also generate high TWC. Storms typically form along CAPE gradients and the CAPE computation is highly sensitive to the boundary layer moisture and temperature profiles that are very challenging for NWP to correctly simulate. Grzych and Mason (2011) analyzed a subset of nearly 100 engine events, and found that only

12% of the events occurred in atmospheres with CAPE greater than 2500 J kg$^{-1}$ (strongly unstable). The vast majority of events occurred in moderate or marginally unstable conditions, and thus CAPE did not appear to be a major driver of events. Due to these issues, the direct use of CAPE in a PHIWC product are likely to be problematic and prone to NWP model-dependent



biases, and lacking in support for correlation to HIWC encounters. Dynamical fields such as those listed above provide forcing for convection, governing where convection does and does not occur, and also modulating storm intensity especially in the case of boundary layer convergence. Unfortunately, NWP models do not always simulate these often narrow convergence zones in the

5    right place and time which would induce error in the PHIWC product. Satellite observations themselves inform the PHIWC product on where deep convection is present and about locations within the convective cloud where storm dynamics are likely to be strongest. Thus it seems that fields such as vorticity and divergence would not provide any new information, and errors in these fields could adversely impact PHIWC product performance.

## 5 Summary

        This paper describes analysis of GEO satellite-derived products relative to in situ TWC observations collected in deep convective clouds by research aircraft during the recent HAIC-

15    HIWC flight campaigns out of Darwin, Australia and Cayenne, French Guiana, and the NASA HIWC-RADAR flight campaign out of Fort Lauderdale, Florida. The intent of this analysis was to determine which satellite-derived products were best for discriminating HIWC conditions depicted by aircraft in-situ total water content (TWC) observations toward development of a Probability of HIWC (PHIWC) diagnostic product. Satellite-derived products such as cloud

20    optical depth (COD), 11 μm IR window brightness temperatures normalized by the tropopause temperature (ΔBTW), the distance to the nearest overshooting cloud top detection in IR or VIS imagery, and distance to the nearest textured region detection in VIS imagery (dOT) can be used to effectively discriminate low-to-moderate from high TWC (i.e. HIWC) conditions. Flights through or beneath optically thick anvil cloud within 40 km of OTs or textured regions were





most likely to experience HIWC conditions. In general, discrimination between moderate and high TWC conditions using satellite-derived signals at cloud top was found to be very challenging because 1) satellites observe signals and processes occurring at or near cloud top while the aircraft sampled cloud conditions typically several kilometers below the top and 2) the

satellite products appear less sensitive to TWC beyond 0.5 g m$^{-3}$ (the 50$^{th}$ percentile of TWC for this satellite-aircraft matched dataset); in other words the same combination of satellite parameters can be present when either 0.5 or 2.0 g m$^{-3}$ TWC are observed.

PHIWC diagnostic products for both day and night application were developed using a fuzzy logic approach based on statistical fits between satellite data and aircraft TWC

observations throughout the three campaigns. Examples from several flights across the three campaigns showed that PHIWC$_{day}$ generally followed TWC trends throughout the flights, with departures being attributed to 1) the fact that the aircraft often sampled far below the cloud top being observed by satellite and 2) the often coarse temporal and spatial resolution of the satellite imagery that cannot resolve rapidly evolving phenomena such as OT signatures. The PHIWC$_{night}$

was able to identify HIWC conditions only when the aircraft flew through or beneath cold cloud tops near OT regions detected in IR imagery. High TWC occurred in one daytime example during Cayenne-2015 beneath optically thick and textured cloud with IR temperatures 25 K warmer than the tropopause. In this case, inclusion of the VIS-based information provided some ability to identify HIWC conditions, where almost no identification would have been possible at

night due to the warm cloud tops.

Validation of the PHIWC products employing a subset of the pixel-scale satellite-aircraft matches, using TWC > 0.5 g m$^{-3}$ to define an event, showed a true and false positive rate (i.e. POD and FAR) of 75% and 35% for PHIWC$_{day}$ with a ROC AUC of 0.75. PHIWC$_{night}$ provided



a 62% POD for an equivalent FAR. The POD rises to 69% in the absence of COD but with inclusion of VIS OT and texture detection in the dOT product. Product performance changed only slightly when TWC > 1.0 g m$^{-3}$ was evaluated. It was found that through the use of TWC vertical profiles retrieved using vertically pointing RASTA cloud radar data on the Falcon-20

aircraft, rather than just in-situ data, a substantial fraction of the false positives in the PHIWC$_{day}$ had some TWC $\geq$ 0.5 g m$^{-3}$ in the column of air above or below the aircraft, reflecting dynamics and/or microphysics that contributed to the high PHIWC. PHIWC values increased on average in the 0.1 to 1 g m$^{-3}$ TWC range and then leveled off at higher TWC values. Very high TWC (> 2 g m$^{-3}$) occurred where the highest PHIWC were identified. PHIWC$_{night}$ for the Cayenne-2015

campaign was significantly lower than the other two campaigns due to the the warmer cloud tops in both an absolute and tropopause-relative sense. Interpreting the product validation is quite challenging due to the fact that satellite observations of cloud top are being used to infer in-cloud processes and microphysical distributions, compounded by satellite sampling limitations such as the coarse 30-minute GOES-13 resolution available during Cayenne-2015.

It is envisioned that the PHIWC diagnostic products could be used in real-time operations in a tactical sense, at up to a one hour lead time, to identify and avoid regions within cloud systems that are likely to generate high TWC. This work demonstrates that the PHIWC products offer improved capability for identifying HIWC conditions in deep convection relative to other known diagnostic products. GEO imager data can now be acquired over a broad (i.e.

hemispheric) geographic domain from a remote server and processed with the OT and texture detection algorithms in ~5 minutes. COD retrievals are more time consuming to produce but inclusion of this product increases accuracy by 8% if some modest latency is not a concern to users. Rapid data access and processing enables real-time production, even with next-generation




satellite imager data such as the GOES-16 Advanced Baseline Imager (ABI, Schmit et al., 2015) and Himawari-8 Advanced Himawari Imager (AHI, Bessho et al., 2016) that feature 4x higher spatial resolution than GOES-13 or MTSAT and have the capability to view hemispheric domains at 10-15 min intervals.

**Data availability**

Satellite imagery and derived products are available from the authors upon request. HAIC-HIWC data (e.g., IKP2, RASTA data) are available solely to core users of the HAIC-HIWC consortium until 14-July-2019. HAIC-HIWC data are available to all other individuals upon

signing a data exchange protocol between 15-July-2019 to 14-July-2022. HAIC-HIWC data are fully publicly available on or after 15-July-2022. HIWC-Radar data are currently available upon request and will be available from the UCAR/NCAR Earth Observing Laboratory:

https://data.eol.ucar.edu/project/545.

**Competing interests**

The authors declare that they have no conflict of interest.

**Acknowledgements**

This work was supported by the Advanced Air Transport Technology Project within the

NASA Aeronautics Research Mission Directorate Advanced Air Vehicles Program. Major North American funding for flight campaigns and associated research was provided by provided by the FAA William Hughes Technical Center and Aviation Weather Research Program, the NASA Aeronautics Research Mission Directorate Aviation Safety Program, the Boeing Co.,





Environment Canada, the National Research Council of Canada, and Transport Canada. Major European campaign and research funding was provided from (i) the European Commission Seventh Framework Program in research, technological development and demonstration under grant agreement n°ACP2-GA-2012-314314, (ii) the European Aviation Safety Agency (EASA)

Research Program under service contract n° EASA.2013.FC27. Further funding was provided by the Ice Crystal Consortium.

We acknowledge the operators of the aircraft that collected the TWC data used in this paper. For the Falcon-20, primary support was provided by the SAFIRE facility for the scientific airborne operations. SAFIRE (http://www.safire.fr), is a joint facility of CNRS, Météo-France

and CNES. For the NASA DC-8, primary support was provided by the NASA Armstrong DC-8 crew, including Tim Moes, NASA DC-8 project manager. We thank the following for HIWC and HAIC project management support: Tom Bond, Jim Riley, and Chris Dumont, Federal Aviation Administration; Ron Colantonio, NASA Aerosciences Evaluation & Test Capabilities Project Manager; Alice Calmels and Fabien Dezitter, Airbus Ind.; Steven Harrah, NASA

Langley's radar principal investigator; Peter May, Rod Potts, Australian Bureau of Meteorology; and Jeanne Mason, the Boeing Company. The authors would like to acknowledge the FAA Aviation Research Division, the NASA Aviation Saftey Program, and Tom Ratvasky and Steven Harrah from NASA for their leadership and support. We also thank the McIDAS Data Center at the University of Wisconsin Space Science and Engineering Center for providing the

archived GEO satellite imagery used in this analysis.



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





| Satellite (and HAIC and/or HIWC Campaign) | Central Wavelength of Spectral Channels | Temporal Resolution | IR (and VIS) Spatial Resolution at Satellite Nadir | Domain Covered By Flights |
|---|---|---|---|---|
| MTSAT-1R (Darwin-2014) | 0.72, 3.8, 6.8, 10.8, 12.0 μm | 10 min | 4 km (1 km) | 20° S - 10° S 120° E - 145° E |
| GOES-13 (Cayenne-2015) | 0.63, 3.9, 6.5, 10.7, 13.3 μm | 30 min | 4 km (1 km) | 3° S - 10° N 46° W - 58° W |
| GOES-13 and GOES-14 (NASA HIWC-RADAR, Ft. Lauderdale Florida 2015) | 0.63, 3.9, 6.5, 10.7, 13.3 μm | 1, 7.5, 15, or 30 mins (Flight Day and Satellite Dependent) | 4 km (1 km) | 14° N - 33° N 55° W – 95° W |

**Table 1: The spectral channels, temporal and spatial resolution, of the two satellite imagers used in this study and the geographic bounds of the study domains.**





| Parameter | $c_1$ | $c_2$ | $c_3$ | max PHIWC threshold | min PHIWC threshold | PHIWC weight |
|---|---|---|---|---|---|---|
| $\Delta$BTW | 0.6953 | 0.9652 | 0.2789 | 0 | 90 | 0.375 (0.700*) |
| $dOT_{VIS+IR}$ | 0.7685 | 0.9411 | 0.3459 | 10 | 1000 | 0.300 |
| $dOT_{IR-only}$ | 0.6595 | 0.9582 | 0.5021 | 10 | 1000 | 0.300* |
| COD | 0.1332 | 0.0063 | | 100 | 0.25 | 0.325 |
| *Indicates weight for IR-only PHIWC | | | | | | |

**Table 2: Coefficients for deriving PHIWC values based on the fits shown by the magenta lines on Fig. 6. ΔBTW, COD, and dOT values exceeding the max PHIWC Threshold are assigned at PHIWC of 1.0. ΔBTW and COD exceeding the min PHIWC Threshold are assigned a PHIWC of 0. Weights for Eq. (4) parameters are indicated in the last column. Weights for Eq. (5) parameters are indicated with an asterisk.**




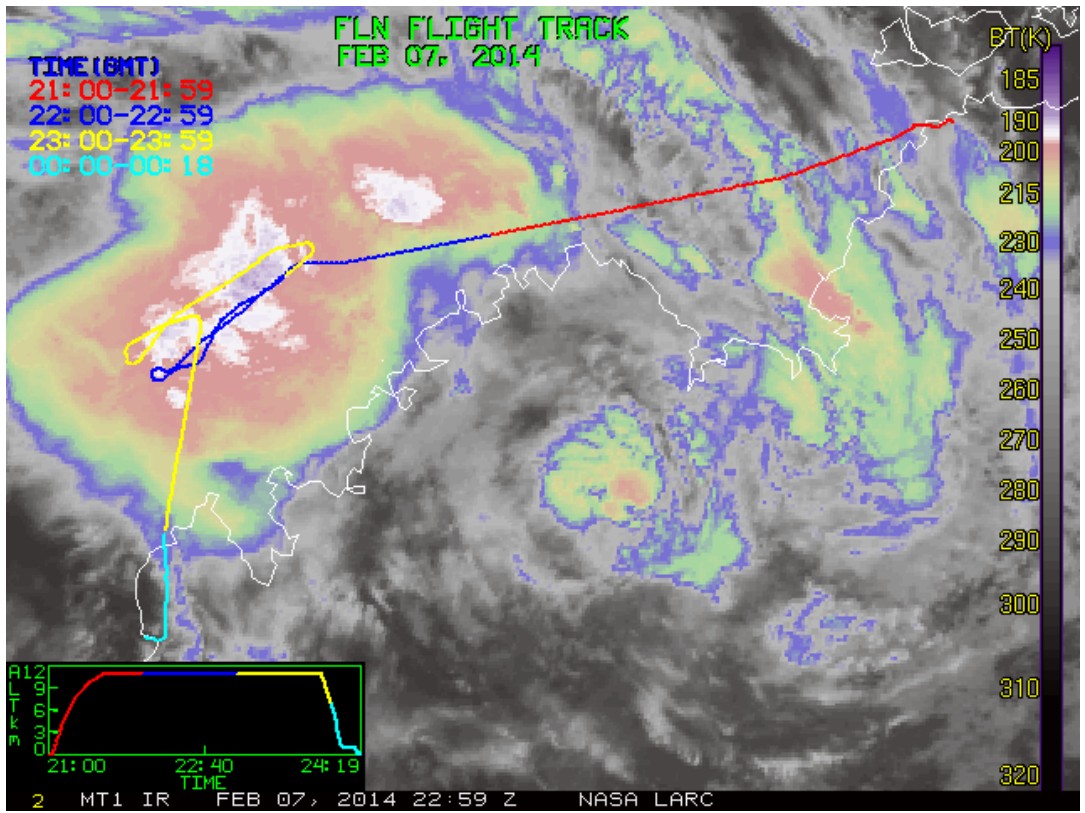

**Figure 1: Flight track of the Falcon-20 on 7 February 2014, which departed from Darwin,**

5     **Australia (red segment) and landed in Broome (cyan segment), overlaid upon a MTSAT-**

**1R JAMI color-enhanced BTW image at 2259 UTC. The track is color coded by the time of**

**observation specified in the upper-left.**





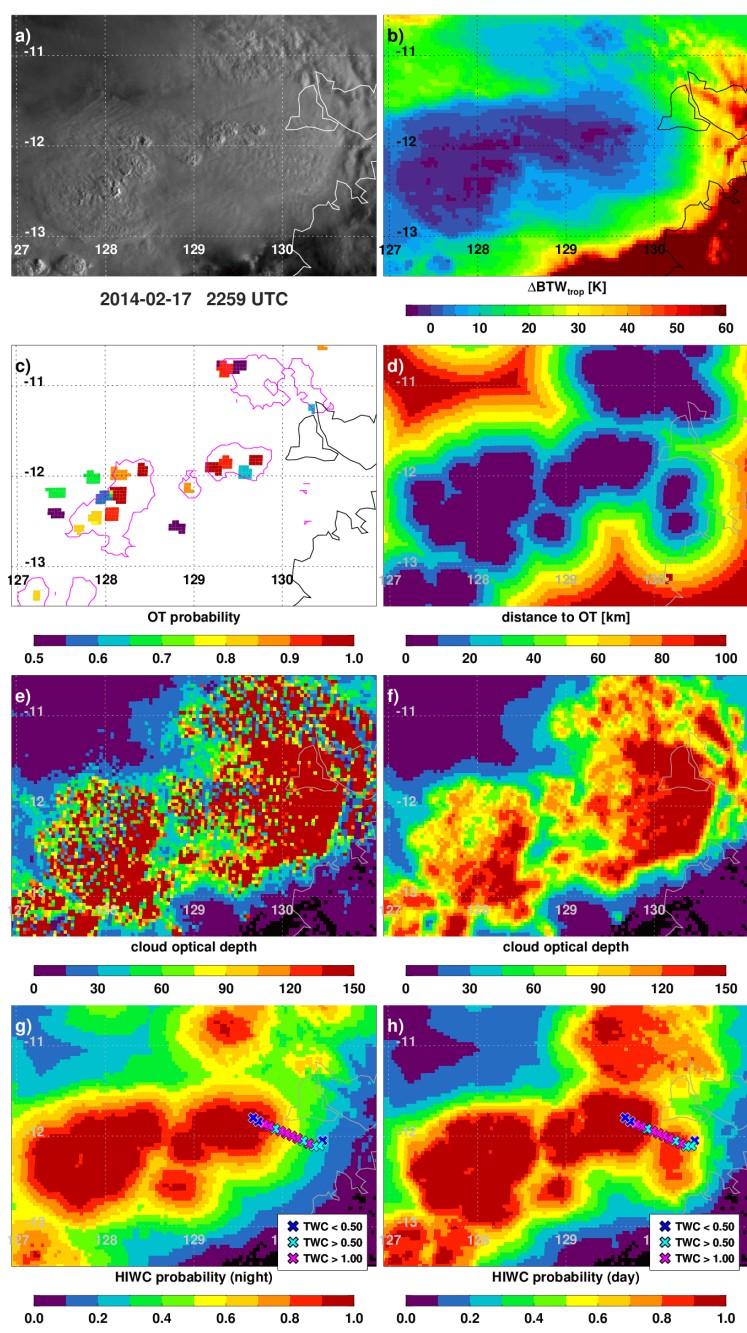

**Figure 2: A series of MTSAT-1R JAMI observations and derived products for an image**

5    **during Falcon-20 Flight 22 of Darwin-2014, timestamped at 2259 UTC but valid over**





Australia at 2302 UTC on 17 February 2014. (a) 0.73 µm VIS reflectance, (b) the difference between the 10.8 µm IR BT and MERRA tropopause temperature (ΔBTW), (c) OT Probability ≥ 0.5 (colored shading) and VIS Rating ≥ 5 (magenta contours), (d) dOT, (e) original unsmoothed COD, (f) smoothed COD, (g) $PHIWC_{night}$ , and (h) $PHIWC_{day}$. Panels

5      (g) and (h) are overlaid with 45-sec mean TWC observations from 2257-2307 UTC.





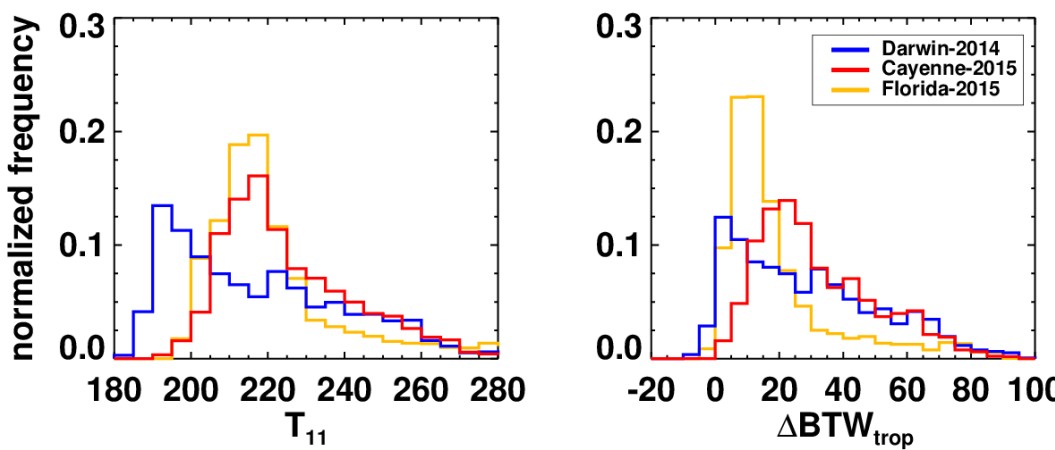

**Figure 3: Distributions of BTW (left) and ΔBTW (right) differentiated by HIWC and HAIC flight campaign using the colors shown in the legend.**



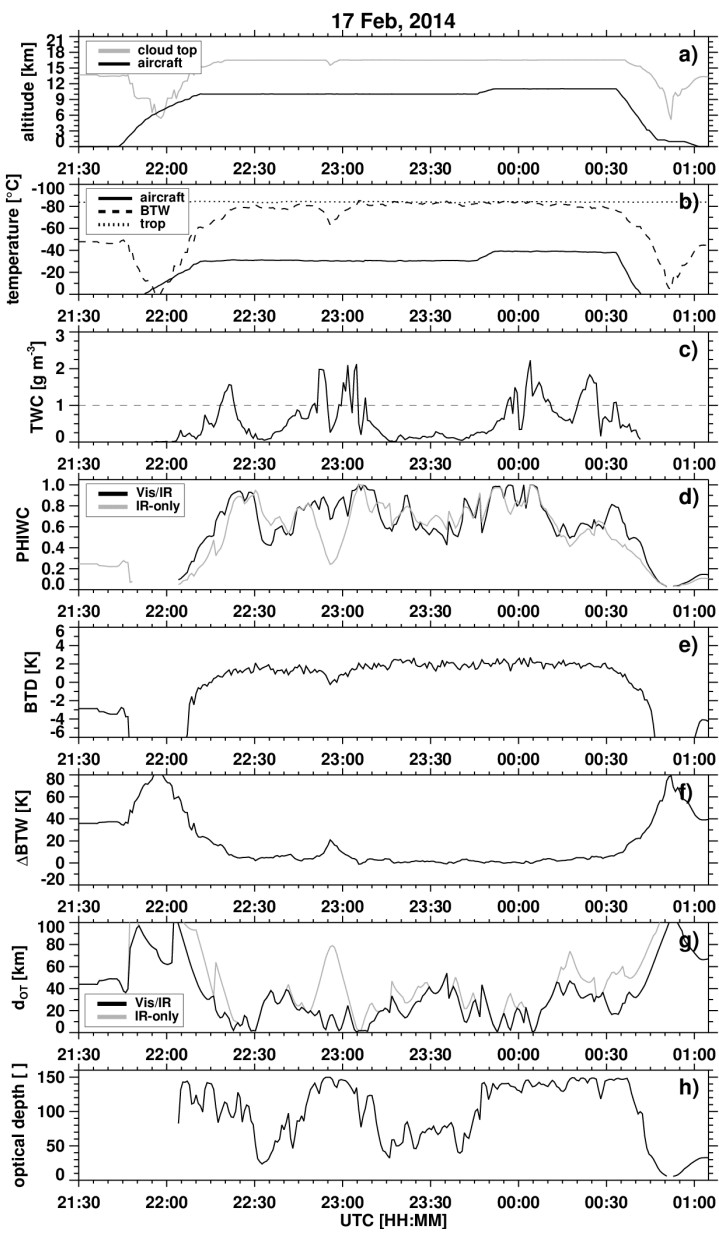

**Figure 4: Time series of matched aircraft and satellite observations for Flight 22 of Darwin-2014 on 17-18 February 2014. (a) Satellite retrieval of cloud top height (grey) and**

5    **the altitude of the Falcon-20 (black). (b) Satellite BTW observations (dashed), MERRA**





**tropopause temperature analysis (dotted), and aircraft static air temperature observations (solid). (c) In-situ 45-second averages of IKP2 TWC measurements. The dashed line indicates TWC = 1 g m$^{-3}$. (d) PHIWC$_{day}$ (black) and PHIWC$_{night}$ (grey), (e) WV-IRW BTD, (f) ΔBTW (g) dOT VIS+IR (black) and IR-only (grey) and (h) smoothed COD. Major and**

5   **minor tick marks represent half-hourly and 5-minute intervals, respectively.**





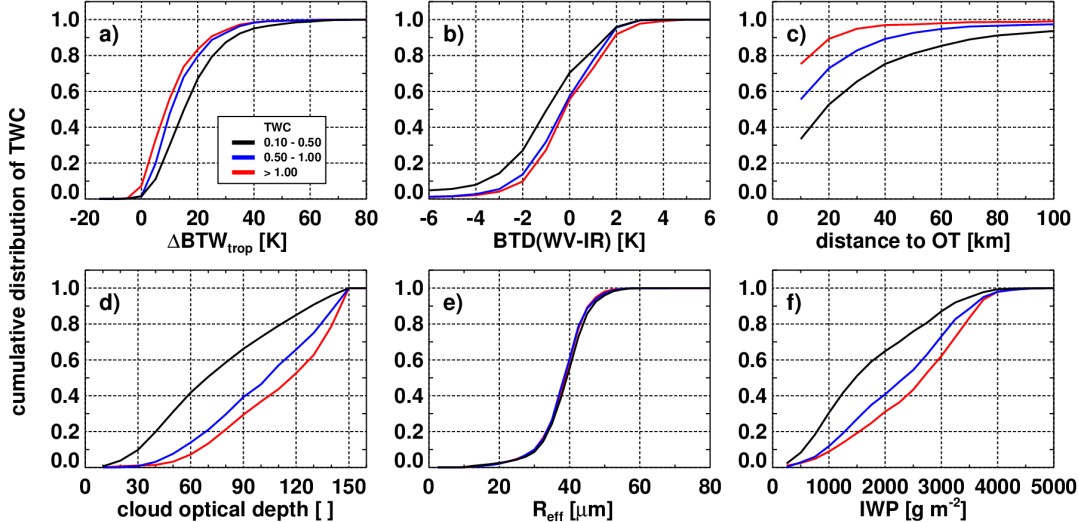

**Figure 5: Cumulative frequency diagrams of satellite-derived a) ΔBTW, b) BTD, c) dOT, d) COD, e) Reff, and f) IWP for the 3 TWC intervals indicated in the legend in panel (a).**



**Figure 6: (left panels) The distribution satellite-derived parameter values as a function of**

**45-sec mean TWC using the training dataset based on 67% of the satellite-aircraft match**



**database (3580 samples). The color represents he density of points in a given region of the scatterplot. The vertical dashed line shows the 0.5 g m$^{-3}$ threshold where satellite parameters start to lose sensitivity to TWC. (right panels) The distribution of TWC as a function of satellite-derived parameters. ΔBTW, COD, dOT VIS+IR, and dOT IR are**

5 **shown from top to bottom. The magenta lines provide a fit to the mean of the distributions and serve as PHIWC fuzzy logic membership functions.**





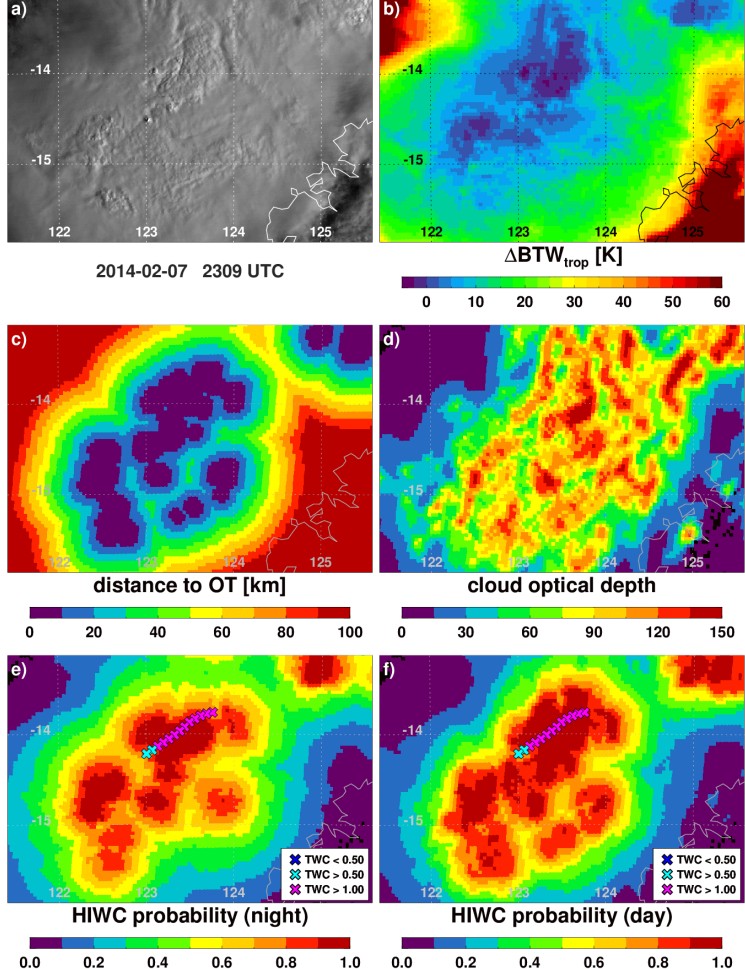

**Figure 7: A series of MTSAT-1R JAMI observations and derived products for an image during Falcon-20 Flight 16, timestamped at 2309 UTC but valid over Australia at 2312 UTC on 7 February 2014. (a) 0.73 µm VIS reflectance, (b) ΔBTW, (c) dOT, (d) smoothed COD, (e) PHIWC$_{night}$, and (f) PHIWC$_{day}$. Panels (e) and (f) are overlaid with 45-sec mean TWC observations from 2307-2317 UTC.**





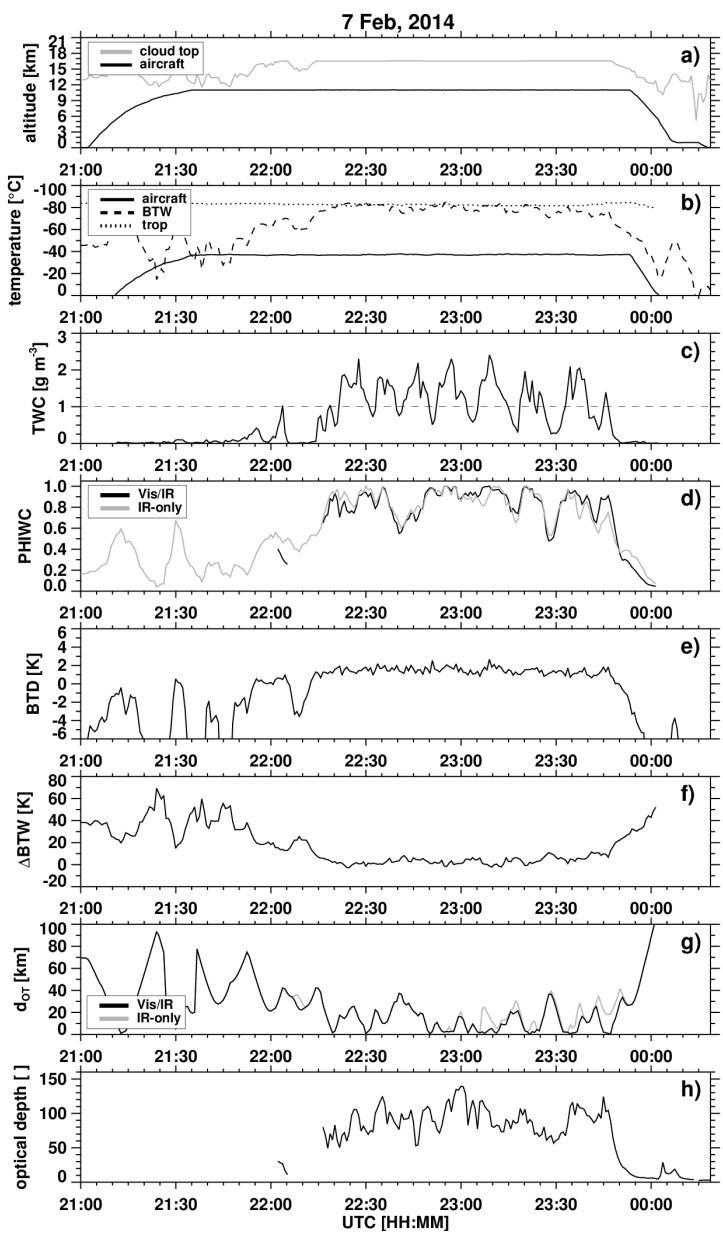

**Figure 8: Time series of matched aircraft and satellite observations for Flight 16 of Darwin-2014 on 7-8 February 2014. The panels are the same as those described for Fig. 4.**





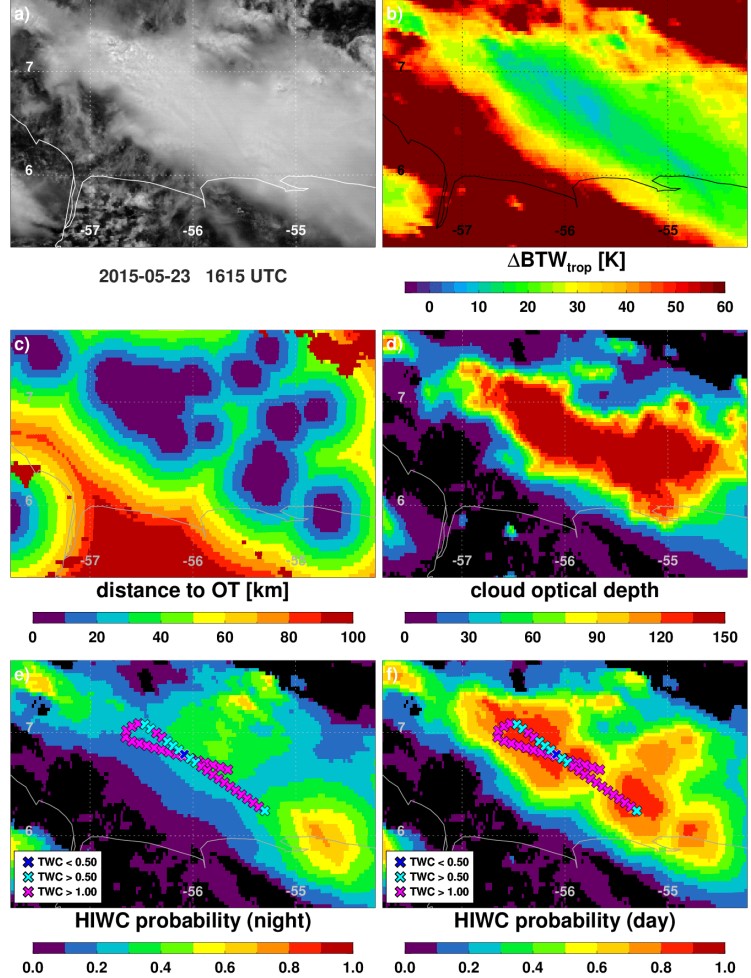

Figure 9: A series of GOES-13 observations and derived products for an image during Falcon-20 Flight 19 of Cayenne-2015, timestamped at 1615 UTC but valid over French Guyana at 1625 UTC on 23 May 2015. The panels are the same as those described for Fig. 7. The 45-sec mean TWC observations are valid from 1615-1635 UTC.





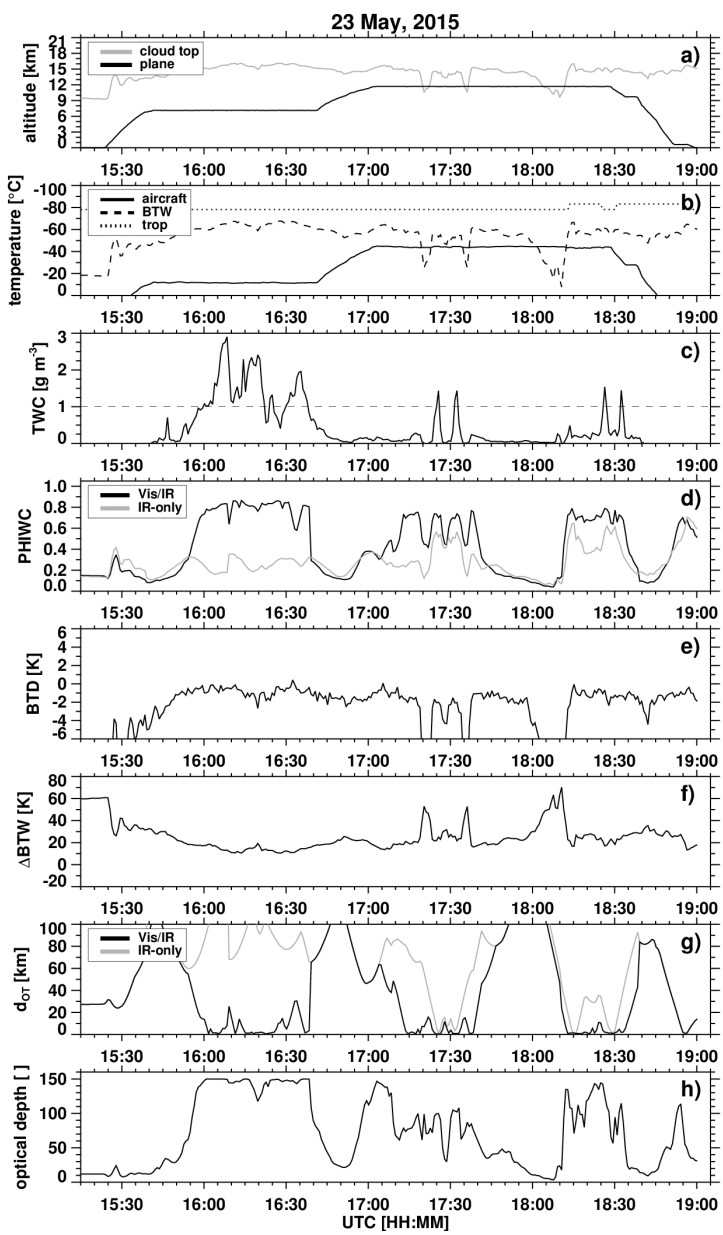

**Figure 10: Time series of matched aircraft and satellite observations for Falcon-20 Flight 19 of Cayenne-2015. The panels are the same as those described for Fig. 5.**



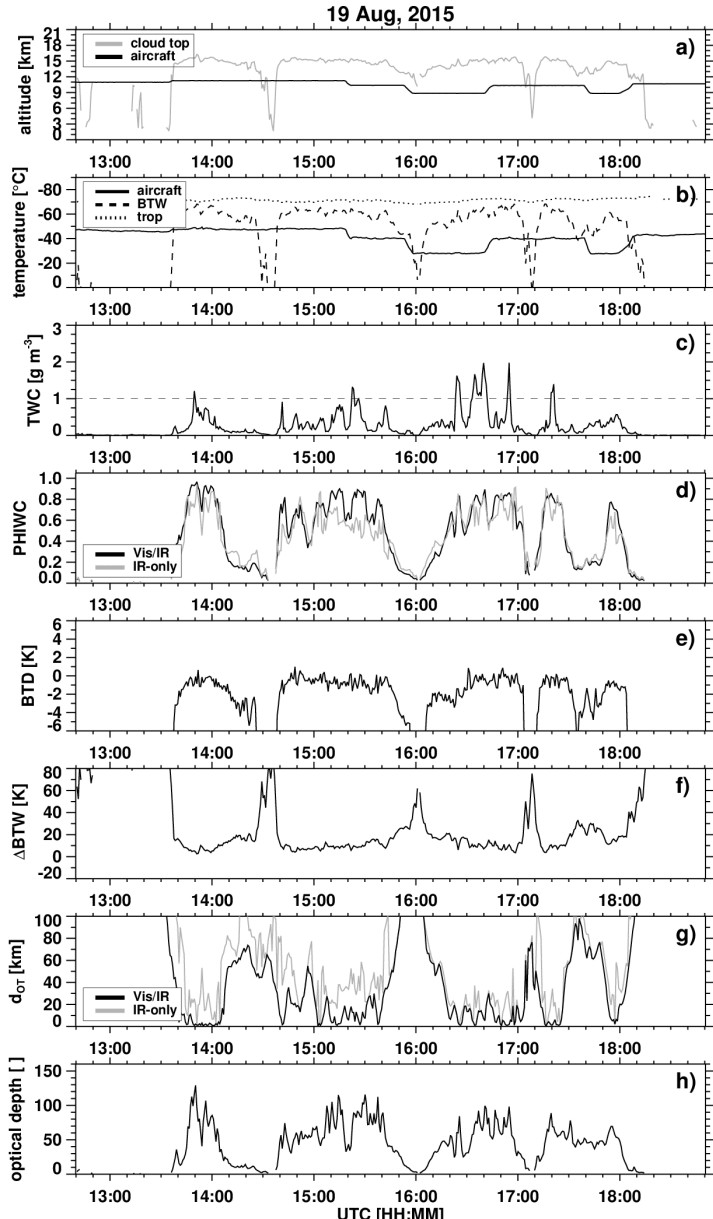

**Figure 11: Time series of matched aircraft and satellite observations for Flight 5 of the Florida-2015 campaign on 19 August 2015. The panels are the same as those described for Fig. 4.**



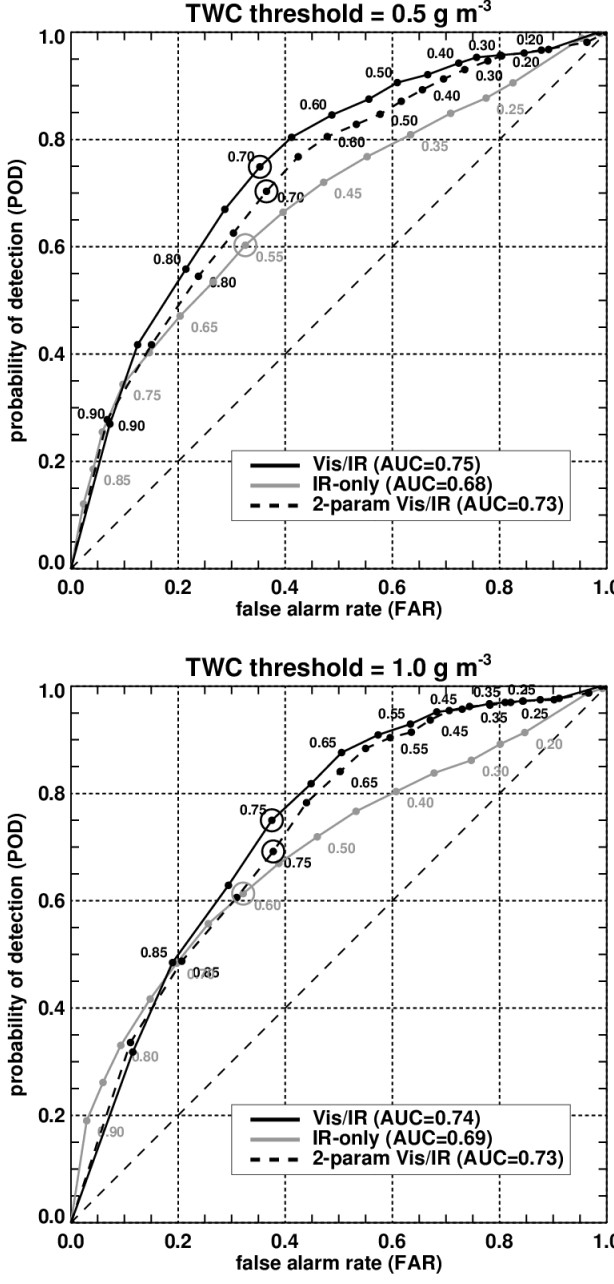

**Figure 12: A receiver operating characteristic (ROC) curve showing the relationship**

5 **between POD and FAR for PHIWC thresholds ranging from 0.0 to 1.0 (labeled at 0.1**





increments) based on TWC thresholds of 0.5 g m$^{-3}$ (top) and 1.0 g m$^{-3}$ (bottom). The solid

black curve represents the three-parameter PHIWC$_{day}$ and the solid grey curve represents

the two-parameter PHIWC$_{night}$. The dashed black curve represents an algorithm

formulation using $\Delta$BTW and dOT analogous to PHIWC$_{night}$, but VIS OT and texture

5    detections in addition to IR OT detections are used to derive dOT. This formulation could

be used for daytime HIWC detection in the event that a COD retrieval product is

unavailable. Optimal PHIWC values based on the maximum area underneath the ROC

curve (AUC) are circled and the corresponding AUC is provided in the legend.



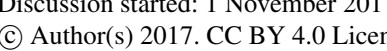

**Figure 13: Box and whisker diagrams showing the relationship between PHIWC$_{day}$ (top,**

5     **N=4598 satellite-aircraft matches), PHIWC$_{night}$ (bottom, N=5371) and TWC. The**

**rectangles show the intraquartile PHIWC range for each TWC bin. The horizontal line**

**within the rectangles shows the median PHIWC. The whiskers show the range of the 1.5%**

**and 98.5% of the distribution and circles are outliers.**