# Peer review of "A Prototype Method for Diagnosing High Ice Water Content Probability Using Satellite Imager Data"

_Atmospheric Measurement Techniques, 2017_

## Referee Comment (RC1) · A. de Laat (Referee) · 22 Nov 2017

Review of AMT(D) paper amt-2017-367.

Yost et al., [2017], A prototype method for diagnosing high ice water content probability using satellite imager data.

This paper describes a satellite observation based method for identification of atmospheric HIWC conditions (now-casting). Identification of HIWC has become a relevant research topic related to aviation, as aircraft have been reported to occasionally suffer from what is known as “ice particle icing” or “in Service icing”, whereby engine damage, engine failure, and loss of control has been reported by aircraft. Such events are thought to be predominantly associated with large concentrations of high altitude small ice crystals, but remains enshrouded in considerable uncertainty about what exactly happens when and where.

The PHIWC product presented in the paper is based on optimizing a set of satellite-based cloud parameters such as cloud top temperatures, optical depths, overshooting tops, and derived quantities.

The product is optimized towards three (aircraft) field campaigns that specifically focused on better characterization of clouds and cloud microphysics for HIWC conditions, thought to be an important condition for the occurrence of “ice particle icing”.

The paper is long but well written, and logically structured. The is a clear buildup of arguments for the parameter choices, detailed analysis of a number of case studies, and a statistical underpinning of the verification statistics of the PHIWC.

Overall, I recommend publication.

Below is listed a number of questions and minor issues.

Jos de Laat

**General comment**

The paper uses both “Total Water Content” and “in situ Total Water Content”. For clarity choose either one, preferably “in situ Total Water Content” in order to avoid confusion as the use of “Total” in other atmospheric research communities often is interpreted to refer to “Total column” . In addition, the abbreviation “TWCi” or “iTWC” could be used, but I leave it up to the authors to decide.

**Specific comments**

- Page 1, line 17. Is that true? High IWC does not necessarily imply high mass concentrations of ice crystals (the reverse obviously does). Maybe rephrase to avoid confusion?

- Page 1, line 20-22. Not directly clear what the “weak reflectivity” refers to. Suggest to change the sentence to:

“... have been document during flight in regions near convective updraft regions that do not appear threatening in onboard weather radar data (weak reflectivity)”

- Page 3, line 5. Suggest to add “... difficult to identify and avoid based on currently available cloud information in the cockpit (mostly weather radar)” or something similar. Reason is that this may change in the future, as evidenced by the satellite data product introduced in the paper.
- Page 4, lines 6-11. There is actually a significant difference between the target of this paper and that of de Laat et al. [2016]. This paper is trying to optimize a set of satellite observed parameters for detection of (local) in-situ TWC exceeding the threshold value of 1 g/m3. De Laat et al. [2016] tries to optimize a set of satellite observed parameters for detection of clouds where anywhere in the vertical in-situ TWC exceeds the threshold value of 1 g/m3. Effectively de Laat et al. [2016] looks first considers the maximum in-situ TWC in a cloud profile, selects those cloud profiles where the maximum in-situ TWC exceeds the threshold, and then optimizes the cloud parameters for detection of this subset. This is fundamentally different from the approach in this paper. For example, in this paper whether in-situ TWC exceeds the threshold elsewhere in the cloud does not matter, whereas in de Laat et al. [2016] it does.

Reason for de Laat et al. [2016] to focus on the maximum in-situ TWC is that weather satellites – which could be useful for an operational service due to their continuous spatio-temporal coverage – only observe clouds from the top down, and only provide either parameters representative for the cloud top, or representative for a (partial/vertically weighted) integrated vertical cloud profile.

This paper and de Laat et al. [2016] thus have fundamentally different goals, which affects identification and characterization of clouds and cloud systems where such conditions occur.

This difference in parameter for which the respective algorithms are designed should be clarified (local in situ TWC vs maximum in situ TWC in vertical cloud profile).

- Page 6, lines 6-10. Discussion of the three field campaigns and whose data is used in the construction of the PHIWC product.

These are three campaigns focusing on particular types of convection, mostly probing active mesoscale convection while also avoiding particular clouds and cloud conditions (see also page 7, lines 15-16; page 8, lines 3-4; lines 16-18; lines 20-22). Although it is accepted that these “ice particle icing” events frequently occur in (mesoscale) convection, they are not exclusively confined to convection alone. This means that the PHIWC product is tuned towards the particular type of convection probed during the field campaigns, while other types of convection or cloud systems are left out. This might result in particular types of convection and cloud systems to be under-sampled and for the PHIWC product to be less accurate in detecting “ice particle icing” conditions associated with types of convection and cloud systems.

Obviously no one knows whether this is really the case, but I think it would be good for the paper to briefly discuss this in the discussion section 4, also because this notion can be translated into some recommendations:

- data sharing by the aviation industry. It would be extremely useful if the aviation industry would be willing to share more data and information about “ice particle icing” events. This is currently not standard practice, which hampers research progress.
- field campaigns focusing on clouds and convective systems not probed during the various HIWC/HAIC campaigns.

- Page 13, lines 10-13. If, as the authors contend, the shadowing effect leads to underestimation of the COD, then I would be tempted to argue that the COD is underestimated (biased low). Smoothing reduces noise, but does not reduce a bias. This is also what is seen in figure 2e-f, where the smoothing reduces the noise. But does that mean that the smoothed field is better?

Shouldn’t the goal be to remove the bias caused by shadowing? If so, that would translate into removing outliers (reduced COD by shadowing) rather than smoothing. For removing outliers other statistical methods should be used.

I don’t believe it is necessary here to change the method for dealing with the shadowing effect (I cannot envision how this could have a significant effect on the verification statistics of the PHIWC product), but if the authors agree, I would recommend mentioning that the smoothing does not remove a bias associated with shadowing, and possibly a statement that this is a topic for future research.

Unless the authors have the possibility to do a quick check on this, in which case results could be included and briefly discussed.

- Page 21, lines 10-11 (section 3.2). I can imagine that the statistics of the occurrence of low/mod/high TWC in relation to their vicinity to OT depends on the type of cloud systems that are sampled during the field campaigns (mesoscale convective systems). Is there any indication that this may be the case?
- Page 37, lines 5-10. Simple question: would the use of motion vectors based on cloud displacement provide a viable possibility for short term prediction? For other applications motion vectors have been shown to allow up to a few hours prediction ahead in time. If the authors think this might be a viable option here as well, they could mention it as something for future considerations.

**Typos**

Page 39, line 1. thermondynamic → thermodynamic

Page 40, line24. thick anvil cloud → thick anvil clouds

---

## Referee Comment (RC2) · Anonymous Referee #3 · 28 Nov 2017

Review of AMT(D) paper  amt-2017-367.

**A Prototype Method for Diagnosing High Ice Water Content Probability Using Satellite Imager Data**, by Yost et al., 2017

This work is the result of analysis of geostationary satellite imagery, together with in-situ total water content (TWC) observations from  three airborne field campaigns, to determine  what  satellite product(s) is (are)  best suited for characterising the ice-water content (HIWC) environment , which may be responsible for the high concentration of ice crystals  sometimes found outside the envelope of altitudes and temperatures normally associated with super-cooled water droplet icing and which may be responsible for  the phenomenon of ice crystal icing, which has resulted in a number of reported incidents affecting the engines of passenger jet aircraft  flying at cruise altitudes of up to 11 km. The authors use a series of satellite derived parameters, such as cloud optical depth, total water content and distance to overshooting tops, visible cloud texture and infrared brightness temperatures among others and matched them in space and time with in-situ airborne measurements of TWC to determine the best combination of these parameters in the process of developing a HIWC satellite diagnostics tool.

In my opinion this is both a very important and comprehensive piece of work. Important because, as it stands, the current fleet of passenger jet aircraft is unfit to deal with HIWC conditions and the attendant threat of ice-crystal engine icing. This means that until a new generation of passenger jet aircraft rolls out of the assembly lines, one which incorporates technologies that will enable aircrews to detect and deal with the threat, the development of a suite of HIWC see-and-avoid capabilities that can be used either directly by aircrews in the cockpit or by air traffic controllers on the ground for tactical re-routing of aircraft operating in the vicinity of HIWC environments and voidance of engine ice crystal icing events, will be the best and possibly only way to avoid the perils posed by ice crystal icing.  The scheme described in this paper is a very good first step towards and a very good example of such capabilities. Comprehensive because the authors have carried out a thorough process of assessing a number of satellite-derived parameters via a comprehensive statistical analysis of their performance in nowcasting the in-situ measured TWC.

I am quite happy to recommend the manuscript for publication in ACP.

What follows is a list of minor comments and recommendations to improve the quality of the manuscript.

- Page 3, line 2:  Reference to Grandin et al, 2014 has no match in the References section's list.

- Whilst not necessarily the main focus of the manuscript, and other than to cite Mason et al's. (2006) hypothesis whereby high mass concentrations of small ice crystals are associated with convective updrafts, the authors offer no explanation or hypothesis on the formation of ice crystals responsible for engine events and their connection to HIWC regions. In my opinion, the Introduction section would benefit from a paragraph or two laying out the possible causes of high-altitude ice-crystal formation and their possible connection to deep convective cloud phenomena, such as jumping cirrus clouds and breaking of gravity waves at anvil altitudes, and cloud features such as overshooting tops, cold-cloud rings and enhanced-V features.

- Page 29, line 14: "...sampled a long-lived but gradually decaying MCS over Louisiana and the offshore over" A noun seems to be missing after "offshore"

- Page 27, line 21. The authors explain the challenges associated with validating the PHIWC product using in-situ TWC measurements, particularly in the case where a decoupling  exists between a cold, optically-thick cloud top with overshooting tops, yielding high PHIWC,  and  low TWC values at low to medium levels, resulting in false detections. While this maybe so, databases of icing events show (see Mason et al., 2006, Bravin et al., 2015), that while the temperatures (altitudes) associated with icing events range from -3° C to -58° C (from 11,000 ft – 45,000ft), the majority of icing events tend to occur either during the cruise and descent phases of flight (the events that occurred during the descent phase of flight had more to do with the susceptibility of jet engines to ice accretion at descent, i.e. low power settings, than with altitude or temperature). Bravin et al. (2015), analyzed eleven events over Japan and Southeast Asia for which 30 minutes (or shorter) satellite imagery was available. Of these, 9 cases occurred during the cruise flight, at a median temperature and altitude of   -44° C and FL 380, respectively. Thus, at least over those areas, there seems to be a prevalence of engine icing events occurring at cruise altitudes and lower temperatures, and while a decoupling between low TWC at lower altitudes and high PHIWC values from cold, optically thick cloud tops, might result in false positives, there is a good chance that the false positives are just not so. Would this warrant considering  weighting the scheme towards optically-thick, cold cloud-tops with overshooting tops yielding high PHIWC even when low TWC are measured at lower altitudes? This should perhaps be mentioned in the text.

- Page 30, line 20: Change "exluded" → "excluded"

- Page 38, line 14: What exactly is "height available"?

- Page 39, line 1: Change "thermondynamic" → "thermodynamic"

---

## Author Comment (AC1) · 23 Jan 2018

**Response to Reviewer Comments (Author Responses in Red)**

**Reviewer 1 - Jos De Laat**

**General comment**
The paper uses both "Total Water Content" and "in situ Total Water Content". For clarity choose either one, preferably "in situ Total Water Content" in order to avoid confusion as the use of "Total" in other atmospheric research communities often is interpreted to refer to "Total column" . In addition, the abbreviation "TWCi" or "iTWC" could be used, but I leave it up to the authors to decide.

We have added in-situ preceding all instances of total water content.  We will keep the acronym TWC, as changing it to TWCi and iTWC would require re-creation of many figures which we do not have the time or motivation to do.

**Specific comments**
• Page 1, line 17. Is that true? High IWC does not necessarily imply high mass concentrations of ice crystals (the reverse obviously does). Maybe rephrase to avoid confusion?

The authors acknowledge a distinction between "ice crystals" and "ice particles".  The term "ice particles" is now used to imply that HIWC conditions are not limited to one type of ice particle (i.e., crystals only) and other types of particles, e.g., hail and graupel, may be present. Rephrased the sentence as followed: "Recent studies have found that ingestion of high mass concentrations of ice particles in regions of deep convective storms, where weak reflectivity in the onboard weather radar data did not indicate a threat, can adversely impact aircraft engine performance. Previous aviation industry studies have used the term high ice water content (HIWC) to define such conditions."

• Page 1, line 20-22. Not directly clear what the "weak reflectivity" refers to. Suggest to change the sentence to:
"… have been document during flight in regions near convective updraft regions that do not appear threatening in onboard weather radar data (weak reflectivity)"

Rephrased the sentence as followed "Recent studies have found that ingestion of high mass concentrations of ice particles in regions of deep convective storms, where weak reflectivity in the onboard weather radar data did not indicate a threat, can adversely impact aircraft engine performance.  Previous aviation industry studies have used the term high ice water content (HIWC) to define such conditions."

• Page 3, line 5. Suggest to add "… difficult to identify and avoid based on currently available cloud information in the cockpit (mostly weather radar)" or something similar. Reason is that this may change in the future, as evidenced by the satellite data product introduced in the paper.

Rephrased the sentence as followed "The microphysical characteristics of HIWC events make this hazard difficult to identify and avoid using only cloud information provided to the cockpit from current weather radars."

• Page 4, lines 6-11. There is actually a significant difference between the target of this paper and that of de Laat et al. [2016]. This paper is trying to optimize a set of satellite observed parameters for detection of (local) in-situ TWC exceeding the threshold value of 1 g/m3. De Laat et al. [2016] tries to optimize a set of satellite observed parameters for detection of clouds where anywhere in the vertical in-situ TWC exceeds the threshold value of 1 g/m3. Effectively de Laat et al. [2016] looks first considers the maximum in-situ TWC in a cloud profile, selects those cloud profiles where the maximum in-situ TWC exceeds the threshold, and then optimizes the cloud parameters for detection of this subset. This is fundamentally different from the approach in this paper. For example, in this paper whether in-situ TWC exceeds the threshold elsewhere in the cloud does not matter, whereas in de Laat et al. [2016[ it does.

Reason for de Laat et al. [2016] to focus on the maximum in-situ TWC is that weather satellites – which could be useful for an operational service due to their continuous spatio-temporal coverage – only observe clouds from the top down, and only provide either parameters representative for the cloud top, or representative for a (partial/vertically weighted) integrated vertical cloud profile.

This paper and de Laat et al. [2016] thus have fundamentally different goals, which affects identification and characterization of clouds and cloud systems where such conditions occur. This difference in parameter for which the respective algorithms are designed should be clarified (local in situ TWC vs maximum in situ TWC in vertical cloud profile).

Thank you for these comments.  We have rephrased to take this and subsequent comments into account:
"One HIWC nowcasting approach has recently been published which seeks to maximize the HIWC event detection rate by identifying any ice cloud with moderate to high cloud optical depth (COD > 20, de Laat et al. 2017).  Their approach seeks to combine satellite-derived cloud parameters to identify regions where TWC could exceed 1 $gm^{-3}$ anywhere throughout the vertical depth of a cloud.  Their binary yes/no HIWC mask indicates where HIWC is possible, but does not provide information on where HIWC is likely especially in deep convection where COD routinely exceeds 20 (Hong et al. 2007). Such an approach could identify a variety of cloud conditions where HIWC may be present, not necessarily restricted to the types of deep convection documented in the Bravin et al. study."

• Page 6, lines 6-10. Discussion of the three field campaigns and whose data is used in the construction of the PHIWC product.

These are three campaigns focusing on particular types of convection, mostly probing active mesoscale convection while also avoiding particular clouds and cloud conditions (see also page 7, lines 15-16; page 8, lines 3-4; lines 16-18; lines 20-22). Although it is accepted that these "ice

particle icing" events frequently occur in (mesoscale) convection, they are not exclusively confined to convection alone. This means that that the PHIWC product is tuned towards the particular type of convection probed during the field campaigns, while other types of convection or cloud systems are left out. This might result in particular types of convection and cloud systems to be under-sampled and for the PHIWC product to be less accurate in detecting "ice particle icing" conditions associated with types of convection and cloud systems. Obviously no one knows whether this is really the case, but I think it would be good for the paper to briefly discuss this in the discussion section 4, also because this notion can be translated into some recommendations:
- data sharing by the aviation industry. It would be extremely useful if the aviation industry would be willing to share more data and information about "ice particle icing" events. This is currently not standard practice, which hampers research progress.
- field campaigns focusing on clouds and convective systems not probed during the various HIWC/HAIC campaigns.

To address your concerns and comments here, we have added the following paragraph to the Discussion section:
"Given that the HIWC and HAIC flight campaigns targeted deep convection in large MCSs, primarily in tropical and sub-tropical regions, the PHIWC product will perform best in identifying icing conditions in such convective cloud environments.  The in-service engine icing events described by Mason et al. (2006), Grzych and Mason (2010), Mason and Grzych (2011), Grzych et al. (2015), and Bravin et al. (2015) occurred in deep convective clouds dominated by large MCSs, but smaller scale convective cloud was also noted. Furthermore, ice particle icing events also include air data probe events, for which there is little published information on cloud type.  It is therefore prudent to note that the PHIWC product may not be tuned for all clouds that would produce significant ice particle icing events.  In order to better understand all types of aircraft icing generated by ice particles, additional data from research aircraft flights through other cloud environments such as smaller scale convective clouds and mid-latitude cyclones would be beneficial.  In addition, provision of detailed information from the aviation industry to the research community on all types of in-service icing events would benefit nowcasting product development in covering the full spectrum of icing conditions."

• Page 13, lines 10-13. If, as the authors contend, the shadowing effect leads to underestimation of the COD, then I would be tempted to argue that the COD is underestimated (biased low). Smoothing reduces noise, but does not reduce a bias. This is also what is seen in figure 2e-f, where the smoothing reduces the noise. But does that mean that the smoothed field is better?

Shouldn't the goal be to remove the bias caused by shadowing? If so, that would translate into removing outliers (reduced COD by shadowing) rather than smoothing. For removing outliers other statistical methods should be used. I don't believe it is necessary here to change the method for dealing with the shadowing effect (I cannot envision how this could have a significant effect on the verification statistics of the PHIWC product), but if the authors agree, I would recommend mentioning that the smoothing does not remove a bias associated with

shadowing, and possibly a statement that this is a topic for future research. Unless the authors have the possibility to do a quick check on this, in which case results could be included and briefly discussed.

These are interesting points. The noise is of greatest concern because application of our approach to a noisy input field will yield a noisy HIWC probability product. It would be difficult to know the size of the noisy areas a priori, so choosing a kernel size to get enough signal in the kernel to identify outliers could be a challenge. Given that our statistical fits are based on the smoothed COD field which incorporates the noisy fields as well as "good" signal via the smoothed product, we are not particularly concerned about the impacts of smoothing on our approach. Note that the noise goes both ways, one can get reflectance enhancements at high solar zenith angle along the cloud sides. So perhaps everything evens out in the end but the smoothing yields a more desirable product for our purposes. We feel that COD retrievals from any group will feature this noise at the pixel scale. These 3-D effects will become more prominent requiring smoothing/filtering in products that use COD as input, especially as visible reflectance is observed at higher spatial resolution (i.e. GOES-16, Himawari-8). We added the following sentence per your request:

"The smoothing process does not remove the COD biases but it provides a more spatially coherent product which is our intent."

• Page 21, lines 10-11 (section 3.2). I can imagine that the statistics of the occurrence of low/mod/high TWC in relation to their vicinity to OT depends on the type of cloud systems that are sampled during the field campaigns (mesoscale convective systems). Is there any indication that this may be the case?

Our opinion is that the updraft depicted as an OT signature is where the vast majority of the ice mass flux is occurring. One can get low or moderate TWC anywhere throughout the cloud but high TWC is concentrated near anvil-penetrating updraft regions, confirmed by studies such as Bravin et al. It is impossible to answer your question for certain because we do not have measurements everywhere in the clouds that we sampled. We have not attempted to classify clouds by type, i.e. mesoscale convective system, air mass convection, organized multi-cell, etc.. so we do not have any indication that our findings vary by storm type.

• Page 37, lines 5-10. Simple question: would the use of motion vectors based on cloud displacement provide a viable possibility for short term prediction? For other applications motion vectors have been shown to allow up to a few hours prediction ahead in time. If the authors think this might be a viable option here as well, they could mention it as something for future considerations.

The atmospheric motion vectors retrieved via cloud tracking represent motions at cloud top. Cell motions within the storm systems could be moving in different directions than the anvil level flow. In deep tropical cloud systems, the aircraft typically do not fly over the storm tops but rather fly through or beneath anvils until they reach storm cores which they hopefully

somehow go around via use of onboard radar. Also, as we note in the paper the cloud conditions an aircraft experiences, especially near anvil, could be due to a number of updraft regions which could be tough to unravel. We noted this as an area of future work.

If you're referring to forward propagating a cloud object's location based on AMVs, you may be able to get the general shape of the anvil to appear in the right vicinity of where it ends up. But the HIWC will be located close to the embedded convective cells in the anvil that are chaotic and hard to predict especially with a few hours lead time as you suggest. Most convective systems are not long lived and organized enough to forward advect for a few hours, and intensification/decay processes along their track (with the exception of perhaps a hurricane) are very hard to predict. So we are not very confident that such a forward propagation would be effective unless you can differentiate cell motions from the motions of the larger system to achieve up to 1 hour prediction.

Nevertheless, it may be useful to identify "outflow channels" from OT regions to narrow down where within the anvil HIWC is more likely to be located. We have added the boldened statement to the section discussing NWP winds to mention this:

It therefore stands to reason that the winds derived from a numerical weather prediction (NWP) model analysis or forecast could be another useful predictor for PHIWC. Unfortunately, outflow from deep convection can alter the upper tropospheric wind environment. Models often do not simulate convection at exactly the right place and time, and even if a storm were accurately simulated, the model may not correctly simulate the interaction of the synoptic scale winds with the convective outflow. These challenges would complicate use of the wind field as a PHIWC predictor in an automated product. **"Mesoscale atmospheric motion vectors" derived from tracking cloud features embedded within anvils using "rapid scan" geostationary imagery (collected at up to 30-second intervals by the GOES-R Advanced Baseline Imager (Schmit et al. 2005)) could help to identify regions where fresh outflow from updrafts is occurring (Bedka and Mecikalski 2005; Apke et al. 2016). Use of observed cloud motions would alleviate concerns about incorrect model depictions of the wind field in deep convection. One could envision that flight through fresh outflow in an anvil would be more likely to experience higher TWC.** Nevertheless, in an environment with multiple updrafts in close proximity to each other, where the aircraft may be upwind of one updraft by several km but downwind of another by tens of km, it would be difficult to understand exactly how, and from where, the observed TWC is generated. Unraveling these complex relationships is a topic for future work.

**Typos**
Page 39, line 1. thermondynamic → thermodynamic
Page 40, line24. thick anvil cloud → thick anvil clouds

Corrected

---

## Author Comment (AC2) · 23 Jan 2018

**Response to Reviewer Comments (Author Responses in Red)**

**Reviewer 2**

This work is the result of analysis of geostationary satellite imagery, together with in-situ total water content (TWC) observations from three airborne field campaigns, to determine what satellite product(s) is (are) best suited for characterising the ice-water content (HIWC) environment , which may be responsible for the high concentration of ice crystals sometimes found outside the envelope of altitudes and temperatures normally associated with super-cooled water droplet icing and which may be responsible for the phenomenon of ice crystal icing, which has resulted in a number of reported incidents affecting the engines of passenger jet aircraft flying at cruise altitudes of up to 11 km. The authors use a series of satellite derived parameters, such as cloud optical depth, total water content and distance to overshooting tops, visible cloud texture and infrared brightness temperatures among others and matched them in space and time with in-situ airborne measurements of TWC to determine the best combination of these parameters in the process of developing a HIWC satellite diagnostics tool.

In my opinion this is both a very important and comprehensive piece of work. Important because, as it stands, the current fleet of passenger jet aircraft is unfit to deal with HIWC conditions and the attendant threat of ice-crystal engine icing. This means that until a new generation of passenger jet aircraft rolls out of the assembly lines, one which incorporates technologies that will enable aircrews to detect and deal with the threat, the development of a suite of HIWC see-and-avoid capabilities that can be used either directly by aircrews in the cockpit or by air traffic controllers on the ground for tactical re-routing of aircraft operating in the vicinity of HIWC environments and voidance of engine ice crystal icing events, will be the best and possibly only way to avoid the perils posed by ice crystal icing. The scheme described in this paper is a very good first step towards and a very good example of such capabilities. Comprehensive because the authors have carried out a thorough process of assessing a number of satellite-derived parameters via a comprehensive statistical analysis of their performance in nowcasting the in-situ measured TWC.

I am quite happy to recommend the manuscript for publication in ACP.

What follows is a list of minor comments and recommendations to improve the quality of the manuscript.

We appreciate your thoughtful summary and kind words.

• Page 3, line 2: Reference to Grandin et al, 2014 has no match in the References section's list.

Corrected

• Whilst not necessarily the main focus of the manuscript, and other than to cite Mason et al's. (2006) hypothesis whereby high mass concentrations of small ice crystals are associated with

convective updrafts, the authors offer no explanation or hypothesis on the formation of ice crystals responsible for engine events and their connection to HIWC regions. In my opinion, the Introduction section would benefit from a paragraph or two laying out the possible causes of high-altitude ice-crystal formation and their possible connection to deep convective cloud phenomena, such as jumping cirrus clouds and breaking of gravity waves at anvil altitudes, and cloud features such as overshooting tops, cold-cloud rings and enhanced-V features.

Because the genesis of HIWC conditions is indeed not a focus of this paper, we prefer to simply cite the work of Mason et al. (2006) on this topic. One reviewer commented that the paper "is long", so we prefer to not add information that isn't vital to the main focus.

• Page 29, line 14: "...sampled a long-lived but gradually decaying MCS over Louisiana and the offshore over" A noun seems to be missing after "offshore"

Added the word "regions" after offshore

• Page 27, line 21. The authors explain the challenges associated with validating the PHIWC product using in-situ TWC measurements, particularly in the case where a decoupling exists between a cold, optically-thick cloud top with overshooting tops, yielding high PHIWC, and low TWC values at low to medium levels, resulting in false detections. While this maybe so, databases of icing events show (see Mason et al., 2006, Bravin et al., 2015), that while the temperatures (altitudes) associated with icing events range from -3° C to -58° C (from 11,000 ft – 45,000ft), the majority of icing events tend to occur either during the cruise and descent phases of flight (the events that occurred during the descent phase of flight had more to do with the susceptibility of jet engines to ice accretion at descent, i.e. low power settings, than with altitude or temperature). Bravin et al. (2015), analyzed eleven events over Japan and Southeast Asia for which 30 minutes (or shorter) satellite imagery was available. Of these, 9 cases occurred during the cruise flight, at a median temperature and altitude of -44° C and FL 380, respectively. Thus, at least over those areas, there seems to be a prevalence of engine icing events occurring at cruise altitudes and lower temperatures, and while a decoupling between low TWC at lower altitudes and high PHIWC values from cold, optically thick cloud tops, might result in false positives, there is a good chance that the false positives are just not so. Would this warrant considering weighting the scheme towards optically-thick, cold cloud-tops with overshooting tops yielding high PHIWC even when low TWC are measured at lower altitudes? This should perhaps be mentioned in the text.

You raise interesting points and we agree that satellite imager radiances and spatial patterns best indicate conditions near cloud top. Only strong vertical motions will noticeably impact the patterns we see. Deeping of the cloud hydrometeors in a column (i.e. convective development from below) will increase cloud optical depth but we cannot really tell what is going on beneath the anvil in a multilayer cloud scene, i.e. a cumulus congestus penetrated by the aircraft with low TWC with anvil over the top from nearby OT producing cell. We're not quite sure what you have in mind when you say "weighting the scheme", but we did add the following underlined sentence that reflects what you mention above near the top of the paragraph in question here.

"A primary reason for disagreement between PHIWC and TWC noted throughout this paper is the fact that the method is attempting to infer conditions at flight level from observations of cloud tops. The PHIWC product best represents conditions within the anvil that aircraft would encounter while cruising at levels typically above 9 km where anvils reside."

• Page 30, line 20: Change "exluded" → "excluded"

Corrected

• Page 38, line 14: What exactly is "height available"?

The text stated "cloud top temperature and height available", so cloud top was attributed to both temp and height. But to minimize confusion we also added cloud top before height.

• Page 39, line 1: Change "thermondynamic" → "thermodynamic"

Corrected as also suggested by Reviewer 1